# Super enhancer regulation of cytokine-induced chemokine production in alcoholic hepatitis

Mengfei Liu[1,12], Sheng Cao [1,12], Li He[2], Jinhang Gao[3], Juan P. Arab [4], Huarui Cui[5], Weixia Xuan[6,7], Yandong Gao[8], Tejasav S. Sehrawat [1], Feda H. Hamdan[1], Meritxell Ventura-Cots[9], Josepmaria Argemi [9], William C. K. Pomerantz [5], Steven A. Johnsen [1], Jeong-Heon Lee[10], Fei Gao[10], Tamas Ordog [1,8,10], Philippe Mathurin[11], Alexander Revzin[8], Ramon Bataller[9], Huihuang Yan [10✉] & Vijay H. Shah [1✉]

Alcoholic hepatitis (AH) is associated with liver neutrophil infiltration through activated cytokine pathways leading to elevated chemokine expression. Super-enhancers are expansive regulatory elements driving augmented gene expression. Here, we explore the mechanistic role of super-enhancers linking cytokine TNFα with chemokine amplification in AH. RNA-seq and histone modification ChIP-seq of human liver explants show upregulation of multiple CXCL chemokines in AH. Liver sinusoidal endothelial cells (LSEC) are identified as an important source of CXCL expression in human liver, regulated by TNFα/NF-κB signaling. A super-enhancer is identified for multiple CXCL genes by multiple approaches. dCas9-KRAB-mediated epigenome editing or pharmacologic inhibition of Bromodomain and Extraterminal (BET) proteins, transcriptional regulators vital to super-enhancer function, decreases chemokine expression in vitro and decreases neutrophil infiltration in murine models of AH. Our findings highlight the role of super-enhancer in propagating inflammatory signaling by inducing chemokine expression and the therapeutic potential of BET inhibition in AH treatment.

[1] Division of Gastroenterology and Hepatology, Mayo Clinic, Rochester, MN, USA. [2] Department of Gastroenterology, Tongji Hospital, Tongji Medical College, Huazhong University of Science and Technology, Wuhan, China. [3] Lab of Gastroenterology and Hepatology, West China Hospital, Sichuan University, Chengdu, China. [4] Department of Gastroenterology and Hepatology, School of Medicine of the Pontificia Universidad Católica de Chile, Santiago, Chile. [5] Department of Chemistry, University of Minnesota, Minneapolis, MN, USA. [6] Department of Respiratory and Critical Care Medicine, Henan Provincial People's Hospital, Zhengzhou, China. [7] Department of Pulmonary and Critical Care Medicine, Mayo Clinic, Rochester, MN, USA. [8] Department of Physiology and Biomedical Engineering, Mayo Clinic, Rochester, MN, USA. [9] Department of Gastroenterology Hepatology and Nutrition, University of Pittsburgh, Pittsburgh, PA, USA. [10] Center for Individualized Medicine, Mayo Clinic, Rochester, MN, USA. [11] University of Lille, Lille, France. [12] These authors contributed equally: Mengfei Liu, Sheng Cao. ✉email: yan.huihuang@mayo.edu; Shah.Vijay@mayo.edu

Alcoholic hepatitis (AH) is a highly morbid condition characterized by acute liver injury in the setting of excess alcohol ingestion. Severe AH can lead to acute-on-chronic liver failure and is associated with a 30-day mortality of greater than 30% with few treatment options[1,2]. Inflammation is a major driver in the pathogenesis of AH, and many cytokines and chemokines are known to be highly upregulated in AH, of which TNFα is a prototypical example. In AH, TNFα exerts transcriptional control over a wide range of cellular functions, leading to intensified inflammation. Multiple chemokines, including *CXCL1* and *CXCL8*, are induced by TNFα, thereby propagating the inflammatory cascade[3]. An influx of immune cells, particularly neutrophils, to the liver leads to cytotoxic injury and inflammation in AH[2,3]. However, the mechanisms of how TNFα signaling leads to downstream transcriptional regulation in AH remain incompletely understood.

Epigenetic mechanisms are increasingly recognized for their critical roles in transcriptional control in human health and disease[4]. These regulatory mechanisms occur at multiple levels of nuclear organization. One example is posttranslational modifications of histone proteins. A variety of modifications, such as acetylation, methylation, ubiquitination, or phosphorylation, can be deposited mainly on the unstructured tails of histone core proteins making up the nucleosomes[4]. The presence or absence of specific histone marks represent signals that regulate the activity of the target gene or regulatory element by directly or indirectly altering local nucleosome function, and editing of these marks can modulate gene expression[5,6]. Indeed, the characterization of genome-wide distribution of histone modifications has provided valuable information about the expression status of genes and the function of regulatory genomic elements. For example, histone H3 trimethylation at lysine 4 (H3K4me3) and acetylation at lysine 27 (H3K27ac) are commonly seen at the promoters of activated genes; whereas trimethylation at H3 lysine 27 (H3K27me3) is typically enriched on repressed genes[7]. Enhancers, which may exert long-range gene transcriptional control through interactions with promoters, a process mediated by 3D chromatin looping and the recruitment and stabilization of transcription factor complexes[8], can be recognized by coincident H3K4me1 and H3K27ac marks, with the latter closely correlating with enhancer activity[9–11]. "Super-enhancers" are clusters of enhancers that are highly enriched for transcription factors and may collectively regulate multiple target genes, often driving a single biological outcome critical for cell type-specific functions[9,12].

The critical function of enhancers in the regulation of TNFα-responsive genes has been demonstrated in many inflammatory processes[13]. However, in spite of recent progress, the role of histone modifications and remote *cis*-regulatory elements in TNFα-mediated transcriptional regulation remains unexplored in AH. Given that activation of inflammatory pathways in AH is global, reversible, and non-mutagenic, we questioned whether epigenetic changes may play a role in the transcriptome changes in AH.

In this work, we utilize RNA-seq and ChIP-seq targeting key histone marks in liver explants of patients with AH and normal controls to characterize the transcriptomic and epigenomic changes in AH. Among the differentially expressed genes, several CXCL chemokines including *CXCL1, 6,* and *8* implicated in neutrophil recruitment show markedly elevated expression in the liver of patients with AH as previously reported[14–16]. These genes are also associated with active histone modifications. We demonstrate that the production of these CXCL chemokines is under the regulation of the TNFα/NF-κB signaling axis. A super-enhancer upstream of the locus containing several CXCL genes is identified and found to orchestrate TNFα-induced upregulation of CXCL chemokines. Both in vitro and in vivo suppression of this super-enhancer, either specifically via epigenome editing or through pharmacological inhibition, decrease the expression of CXCL chemokines and limit neutrophil infiltration into liver tissue. Functional studies on this CXCL super-enhancer highlight its role in the propagation of inflammatory signals in AH. Understanding the roles of distal *cis*-regulatory elements and corresponding epigenetic regulation in AH is not only crucial for advancing knowledge about disease pathogenesis, but also may open new mechanistic avenues for the development of novel therapeutics generalizable to other inflammatory diseases in the liver and in other organ systems.

## Results

**RNA-seq and histone modification ChIP-seq demonstrate differential gene expression and histone modification patterns in AH and normal livers.** To characterize the gene expression changes in the livers of patients with AH, liver explants were obtained at the time of liver transplantation from patients with severe AH and analyzed by RNA-seq. Individuals with normal livers were used as controls. Demographics and clinical characteristics of AH subjects are summarized in Supp. Fig. 1. Principal component analysis (PCA) of RNA-seq data revealed profound transcriptomic alterations in AH livers (Suppl. Fig. 2a). We identified 950 genes that are significantly upregulated and another 761 genes significantly downregulated in AH livers (FDR ≤ 0.01 and $\log_2$ (fold change) ≥ 1.5, Suppl. Fig. 3a). To understand whether epigenetic mechanisms play a role in the transcriptional dysregulation in AH, we also performed ChIP-seq of four important histone modifications on these samples, including H3K4me3 preferentially associated with promoters, H3K4me1 and H3K27ac preferentially associated with enhancers, as well as H3K27me3 associated with Polycomb repression. Peaks were mapped to the nearest transcription start site (TSS). Plotting the ChIP-seq signal density over TSS ± 2 kb regions revealed that AH upregulated genes are generally associated with increased signals of active marks (H3K4me3, H3K4me1, and H3K27ac) and decreased signal of H3K27me3, and a reverse trend was observed for AH downregulated genes (Suppl. Figs. 3a, 4). A similar pattern was revealed when plotting the aggregate histone mark signals (reads per million) over TSS ± 5 kb regions of AH up- and downregulated genes (Suppl. Fig. 3b). These findings suggest that histone modification patterns correlate with gene expression in a significant subset of genes dysregulated in AH and further raise the possibility that epigenetic mechanisms may play regulatory roles in the disease process.

Integration of differential occupancy profiles of histone modifications with differential gene expression identified candidate genes that may be subjected to epigenetic regulation (schema shown in Fig. 1a). In this analysis, 396 of the 950 significantly upregulated genes and 512 of the 761 significantly downregulated genes in AH livers also showed corresponding statistically significant differential histone mark occupancy (Fig. 1b). Ingenuity pathway analysis (IPA) identified enrichment of numerous signaling pathways associated with differential expression and histone mark profiles in AH. The top affected pathways included granulocyte and agranulocyte adhesion and diapedesis pathways, and the former was the second most affected pathway when only the differentially upregulated genes with congruent histone modifications were examined (Fig. 1c). Indeed, PCA of the differentially expressed genes with congruent histone modifications as well as the granulocyte/agranulocyte adhesion and diapedesis pathway associated genes showed excellent discrimination between AH and normal samples (Suppl. Fig. 2b, c). Within the granulocyte/agranulocyte adhesion and diapedesis

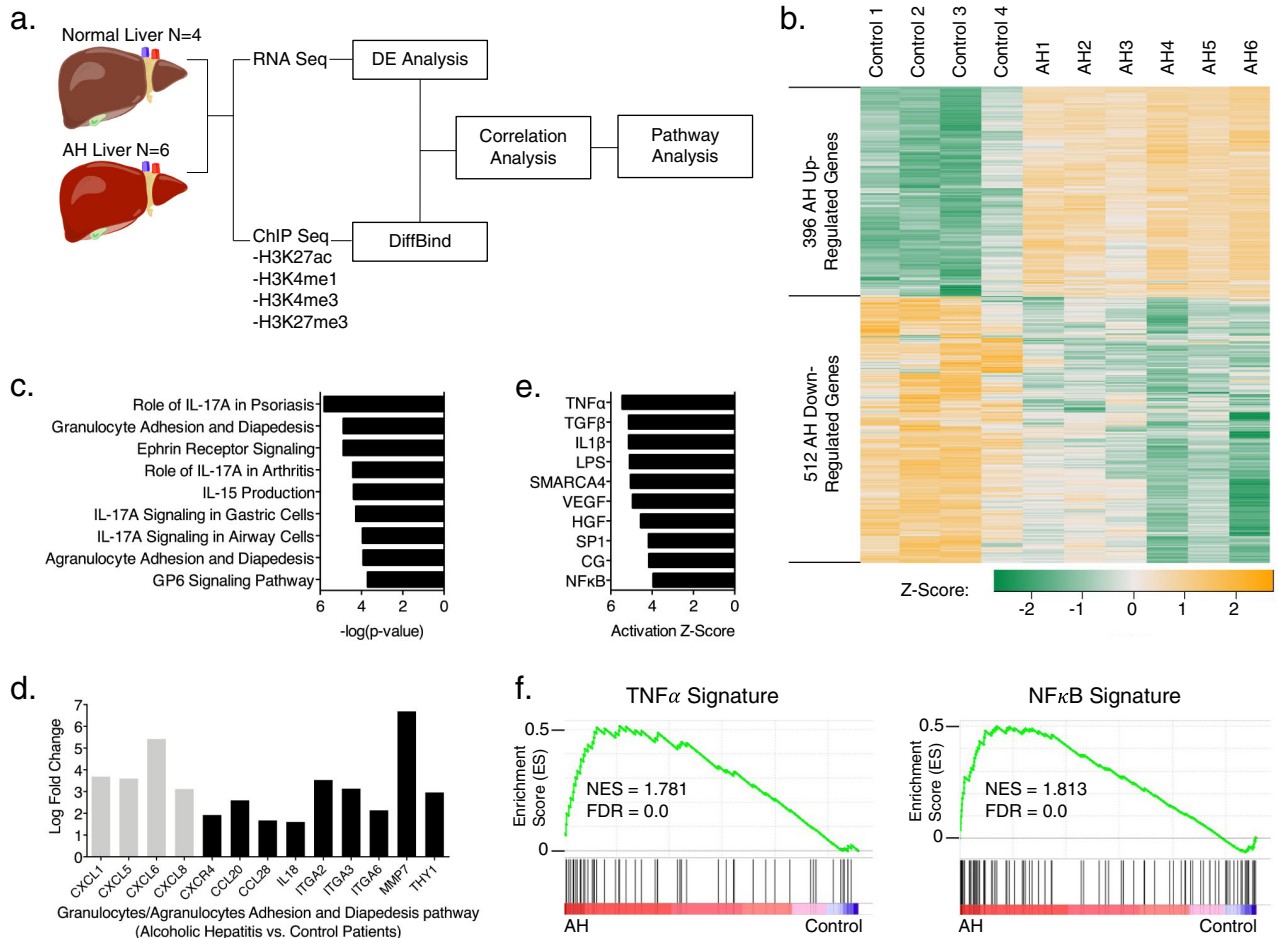

**Fig. 1 RNA-seq and histone mark ChIP-seq of AH and normal livers show significant differences. a** Schematic of RNA-seq and ChIP-seq analyses pipeline. **b** Heatmap of differentially expressed genes from the integrated analysis of RNA-seq and ChIP-seq (gold: upregulated genes; green: downregulated genes). **c** IPA of upregulated genes from the integrated analysis. Top ten affected canonical pathways are listed along with their respective inverse log of *p* values. Right-tailed Fisher's Exact Test was used for the calculation of *p* values. **d** Differentially expressed genes from the granulocytes/ agranulocytes adhesion and diapedesis pathways are listed. Four CXCL chemokines 1, 5, 6, and 8 are located at the same locus and are colored light gray. **e** Upstream regulator analysis from IPA. Top ten activated upstream regulators are listed along with their respective normalized *z*-scores. **f** GSEA of TNFα and NF-κB pathway target genes. AH enriched genes are plotted to the left and control enriched genes are plotted to the right. Normalized enrichment score (NES) and false discovery rate (FDR) are listed for the analyses.

pathways, several CXCL chemokines were located near the same gene locus on chromosome 4 and showed remarkable upregulation in patients with AH (Fig. 1d). We elected to focus our subsequent analysis on *CXCL1, 6*, and *8* due to their robust expression in our AH cohort and their critical role in immune cell chemotaxis, particularly neutrophils[15,17]. These genes were enriched for the active modifications (H3K27ac and H3K4me3) and depleted of repressive mark H3K27me3 within both promoter regions and gene bodies in AH (Suppl. Fig. 5). Given the central role of neutrophilic infiltration in the pathogenesis of AH, the upregulation of these CXCL genes provides a mechanistic link between local inflammation and systemic neutrophilic mobilization in the development of AH. To study the regulatory mechanism driving gene expression reprogramming in AH, upstream regulator analysis was performed and identified multiple pathways well-studied in liver inflammatory signaling, including TNFα, TGFβ, IL1β, among others (Fig. 1e). Most notably, the TNFα/NF-κB pathway has previously been identified to be an upstream regulator of chemotaxis genes[18]. Indeed, gene set enrichment analysis (GSEA) of TNFα and NF-κB pathway genes showed selective upregulation in patients with AH (Fig. 1f). Upstream regulator analysis also highlighted multiple epigenetic

modifiers, including SMARCA4 and SP1, that are activated in AH (Fig. 1e). SMARCA4 regulates accessibility to the chromatin[19], and SP1 interacts with histone-modifying enzymes to affect gene expression[20]. The upregulation of epigenetic modifier pathways supports a role for epigenetic regulation in AH.

**Liver sinusoidal endothelial cells are a major source of CXCL chemokines in AH downstream of the TNFα/NF-κB signaling axis.** The coincidence of mutually exclusive marks H3K27ac and H3K27me3 over the same gene promoters (Suppl. Fig. 5) suggested concordant epigenetic events occurring in different cells— possibly, different cell types. Therefore, we set out to identify the cellular source of elevated liver CXCL chemokine production in AH. Currently, the sources of various CXCL chemokines in normal and diseased livers remain ambiguous. Immune cells are well-known sources of chemokine production, but liver resident cells have also been suggested as a source of abundant chemokine production under stimulation[21–24]. Given their residency and relative abundance in the liver, liver resident cells may have unique roles in sensing and propagating inflammatory signals that result in immune cell recruitment. Examining the publicly

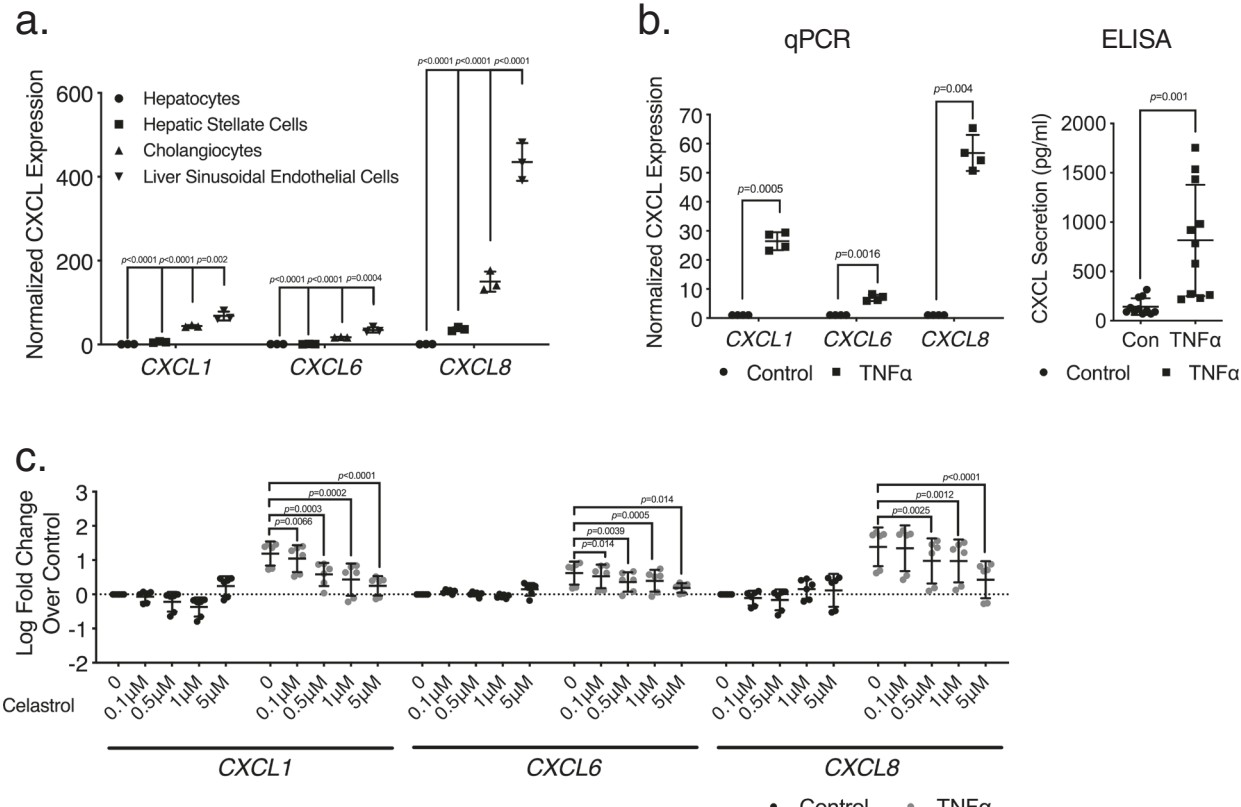

**Fig. 2 LSECs are a major source of CXCL chemokines in the liver under the control of TNFα/NF-κB signaling. a** RNA-seq of normal liver cell types. Expression levels of CXCL genes (expressed in RPKM) were normalized (see supplemental methods for normalization details) and ratios are plotted. Data represented as mean ± SD. One-way ANOVA analysis was performed, with post hoc Dunnett's multiple comparison correction ($n = 3$ biologically independent samples for each cell type). **b** TNFα significantly increased expression of CXCL chemokines revealed by qPCR ($n = 4$, biologically independent samples from two independent experiments, experiments repeated more than five times with similar results) and by ELISA ($n = 11$, biologically independent samples from three independent experiments). CXCL chemokines expression is shown as fold change over control condition for qPCR and as pg/ml by ELISA. Data represented as mean ± SD. A two-sided, paired $t$-test analysis was performed. **c** LSECs were pretreated with varying amounts of Celastrol (0 to 5 μM) and exposed to medium with or without TNFα. qPCRs were performed for quantifying the expression of *CXCL1, 6*, and *8*, and shown in $\log_{10}$ fold over basal expression. Data represented as mean ± SD. One-way matched-pairs ANOVA analysis was performed with post hoc Dunnett's multiple comparison correction ($n = 6$, biologically independent samples from three independent experiments). There was a significant linear trend of decreasing *CXCL1, 6, 8* expressions with increasing Celastrol concentrations with TNFα ($p < 0.0001$ for all groups).

accessible database for human gene expression profiles published by the FANTOM5 Consortium identified liver sinusoid endothelial cells (LSECs) as a predominant cell type in normal liver tissue that expresses *CXCL1, 6*, and *8* (Suppl. Fig. 6). To confirm this finding, we analyzed RNA-seq data in several primary human liver cell types, including LSECs, intrahepatic biliary epithelial cells (cholangiocytes), and hepatic stellate cells, as well as hepatocyte cell-line HepG2, and found LSECs to have the highest expression of *CXCL1, 6*, and *8* (Fig. 2a). We also analyzed recent single-cell RNA-seq (scRNA-seq) data in human and mouse livers. We found cholangiocytes, macrophages, and LSECs to be among the top sources of CXCL chemokine production in multiple studies (Suppl. Table 1)[25,26]. Endothelial cells secrete soluble factors to influence the behavior of neighboring cells in a process termed angiocrine signaling, but the signaling and regulatory mechanisms underlying this process remain largely unknown. As LSECs represent a large cell population in the liver, we hypothesize that LSECs, as resident liver cells with direct contact with infiltrating immune cells, may be particularly important in the early inflammatory response before the onset of significant immune cell infiltration. Therefore, further mechanistic studies were performed on endothelial cells. As TNFα is a well-known activator of chemotaxis and was identified as a key upstream

regulator of differentially expressed genes in our pathway analysis above (Fig. 1e), we examined the LSEC response to TNFα simulation. LSECs demonstrated a significant increase in the expression of CXCL genes upon exposure to TNFα in vitro (Fig. 2b). NF-κB was another key signaling intermediary in our upstream regulator analysis (Fig. 1e). Using the transcription factor binding profiles curated in the JASPAR database, we identified consensus NF-κB binding sites (>95% consensus) in the promoter region of each CXCL chemokine at this locus (Suppl. Fig. 7). To test the importance of NF-κB signaling in LSECs, Celastrol, a proteasome inhibitor that blocks NF-κB transport to the nucleus[27], was applied to LSECs, which significantly decreased CXCL chemokine expression and abrogated the stimulatory effect of TNFα stimulation in a dose-dependent manner (Fig. 2c). These findings suggest that LSEC production of CXCL chemokines is NF-κB dependent, and TNFα augments CXCL expression through the action of NF-κB. To simulate the process of neutrophil recruitment in vitro, we utilized an LSEC-coated microfluidic system and a Transwell two-chamber system to assess the effect of CXCL chemokines on neutrophil adhesion and chemotaxis. Recombinant *CXCL1* increased neutrophil adhesion as expected. Conditioned medium from LSECs increased neutrophil attachment compared to fresh medium, likely by

increasing secretion of neutrophil chemotaxis activators, such as CXCL chemokines, as Girbl et al. previously demonstrated (Suppl. Fig. 8)[28]. This chemotactic effect was further accentuated by TNFα pretreatment of LSECs and decreased with pretreatment with Celastrol, illustrating that observed in vivo CXCL differential expression has biologic relevance (Suppl. Fig. 9).

**Super-enhancer-induced expression of multiple CXCL chemokines is NF-κB-dependent.** Enhancer–promoter interactions play an important role in the control of gene transcription. A distal enhancer simultaneously modulating the expression of multiple genes encoding CXCL chemokines has been identified by high-throughput screening but not functionally verified until recently in HUVEC cells[29,30]. We hypothesized the presence of a super-enhancer regulating the expression of chemokines CXCL1, 6, and 8 under the control of TNFα, and utilized circular chromosome conformation capture-sequencing (4C-Seq) as a discovery tool to identify this putative CXCL master regulatory element in LSEC cells. 4C-Seq detects regions showing strong chromatin interactions with the region of interest (or bait region). Here, we used the CXCL1 promoter as the "view point" and investigated the three-dimensional interactions occurring genome-wide with this region in the presence or absence of TNFα stimulation. A 75 kb region proximal to the CXCL8 gene locus was highly associated with the CXCL1 promoter following TNFα stimulation, particularly in response to TNFα (Fig. 3a). In AH livers, in response to TNFα, this gene locus showed increased occupancy of H3K27ac and H3K4me1, which together define active enhancer sites (Fig. 3b). In contrast, the same site showed much-reduced interactions with the CXCL1 promoter in HEK293T cells which expresses CXCL minimally (Suppl. Fig. 10a). To further study the role of NF-κB in LSEC CXCL expression, we utilized NF-κB (RELA subunit) ChIP-seq datasets in HUVEC cells (GSE53998) and analyzed NF-κB binding in the CXCL locus. Following TNFα stimulation, multiple strong NF-κB peaks were found in the putative enhancer region as well as in the promoter regions of CXCL genes, showing NF-κB binding motif curated in the JASPAR database (Suppl. Fig. 10b). We performed NF-κB (RELA/p65) ChIP-seq in LSECs and also observed enhanced NF-κB binding with TNFα stimulation (Suppl. Fig. 11a). The discrepancy of the number and sizes of peaks between HUVEC/LSEC ChIP-seq might be due to endothelial cells from different origins or the usage of different anti-NF-κB antibodies in this assay. ChIP-qPCR experiments further confirmed that NF-κB binding was increased in the promoters of multiple CXCL genes in response to TNFα stimulation (Suppl. Fig. 12a). Bromodomain and extraterminal domain-containing protein 4 (BRD4) is a transcriptional regulator known to bind to super-enhancers and drive super-enhancer function[13]. BRD4 binding was strongly enriched in the putative CXCL enhancer region in HUVEC cells after TNFα stimulation, largely overlapping with NF-κB peaks (Suppl. Fig. 10b) (GSE53998). We examined the binding of NF-κB to the predicted binding sites in the putative enhancer region in LSEC cells. These sites were selected based on the enrichment of the H3K27ac ChIP-seq signal in AH livers as well as strong NF-κB binding in HUVEC and LSEC cells (Suppl. Fig. 10b, Suppl. Fig. 11a). The site with the highest NF-κB ChIP-seq signal also showed the most TNFα responsiveness in ChIP-qPCR (Suppl. Figs. 11a, 12a). Indeed, both NF-κB and BRD4 bound specifically and robustly to this site after TNFα stimulation (Fig. 3c). Given the large size of this putative enhancer region and high occupancy of the transcriptional coactivator BRD4, we hypothesized that our aforementioned interaction site identified through 4C-Seq may be a super-enhancer with CXCL1 as one of its target genes in LSECs.

Computationally, super-enhancers are typically defined based on strong enrichment of H3K27ac, or alternatively, coactivators BRD4 or the Mediator complex subunit 1 (MED1), using the Rank Ordering of Super Enhancer (ROSE) algorithm[31]. However, likely due to the fact that our AH liver ChIP-seq data was from a mixture of various cell types, the relative H3K27ac intensity of this broad region did not reach the signal cutoff threshold for super-enhancers in the ROSE algorithm. We performed H3K27ac ChIP-seq on LSEC with and without TNFα stimulation. H3K27ac occupancy over the putative CXCL super-enhancer was increased with TNFα treatment. Indeed, ROSE identified this putative enhancer as a super-enhancer in LSEC cells both in the presence and absence of TNFα, but its site ranking rose higher under TNFα stimulation (Fig. 3d, e). ROSE analysis of HUVEC H3K27ac ChIP-seq data (GSE53998) also identified this putative enhancer as a super-enhancer in HUVEC cells (Suppl. Fig. 11b). To explore whether this super-enhancer also regulates other CXCL genes at this gene locus in LSEC, we performed a targeted chromosome conformation capture (3 C) experiment to examine the chromatin interaction between this super-enhancer and the promoters of CXCL1, 2, 3, 5, 6, or 8 genes. Sequences from a nearby noninflammatory gene, RASSF6, as well as intronic segments between CXCL genes, were used as negative controls (Fig. 3f). The previously selected NF-κB binding site on the super-enhancer was chosen as the reference site to assess its interactions with various promoters. We found increased interactions between the super-enhancer with promoters of CXCL1, 2, 3, 6, and 8, which increased with TNFα stimulation (Fig. 3f and Suppl. Fig. 13). CXCL5 did not show enriched interaction with the putative super-enhancer. Collectively, these chromatin binding and interaction experiments identified a super-enhancer with multiple CXCL chemokines as putative target genes in LSECs.

**Epigenetic suppression of the CXCL super-enhancer and CXCL1 promoter sites modulates chemokine gene expression.** Next, we aimed to demonstrate that targeted suppression of this super-enhancer at strategic sites inhibits CXCL gene expression. Using an endonuclease-deficient Cas9 protein (dCas9) fused with the Krüppel associated box (KRAB) domain, which recruits histone deacetylases and methyltransferases to target genes[32], we introduced targeted, epigenetic suppression in the super-enhancer region in LSECs (Fig. 4a). Single guide-RNAs (sgRNA), which dictate site-specificity of dCas9-KRAB, were designed to target the NF-κB binding sites on the super-enhancer described above, and the sgRNA with the strongest effect compared to empty sgRNA vector in dCas9-KRAB transduced cells was selected for subsequent experiments (Suppl. Fig. 14). We found that dCas9-KRAB significantly reduced expression of multiple CXCLs without significant cytotoxicity (Suppl. Fig. 15), while the expression of nearby noninflammatory gene MTHFD2L was unchanged (Fig. 4b). CXCL1, 6, and 8 appeared to be more sensitive to dCas9-KRAB-mediated suppression of the CXCL super-enhancer compared to CXCL 2, 3, and 5 (Suppl. Fig. 16a). We also performed site-specific gene repression targeting a predicted NF-κB binding site in the promoter of CXCL1. As expected, CXCL1 expression was suppressed, but other CXCL genes and the nearby noninflammatory gene were unaffected (Fig. 4c and Supp. Fig. 16b). Cells receiving sgRNA treatment only without dCas9-KRAB showed no change in expression of CXCL genes (Suppl. Fig. 16c, d). To demonstrate that the effect of dCas9-KRAB was target-specific, we examined the enrichment of the repressive mark H3K9me3, which is thought to be the main mechanism of KRAB-mediated epigenetic silencing. H3K9me3 occupancy at the targeted NF-κB binding sites in the super-enhancer and CXCL1 promoter regions was increased with dCas9-KRAB-mediated

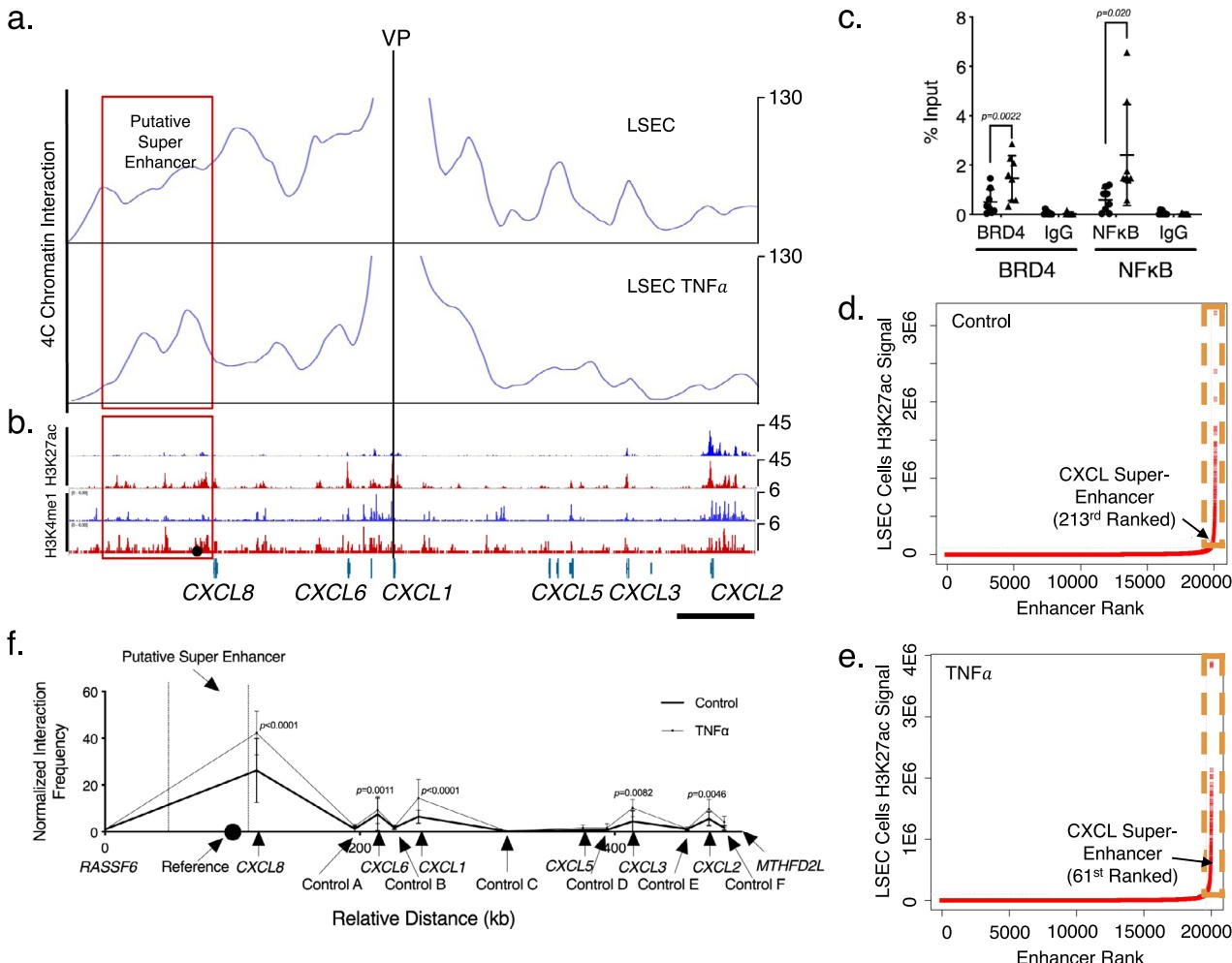

**Fig. 3 Identification of a super-enhancer for CXCL chemokines. a** 4 C was performed on LSEC cells with and without TNFα stimulation. Interactions with the *CXCL1* promoter were plotted using read counts. A genome region of about 75 kb contained two peaks of *CXCL1* interaction under TNFα stimulation (red box). Viewpoint (VP) was labeled with a blackline indicating the location of the reference sequence. **b** H3K27ac and H3K4me1 ChIP-seq signals of normal (blue) and AH (red) livers were plotted. The black dot indicates the location of the NF-κB site (E1) targeted for subsequent analyses. Scale bar represents 50 kb. **c** ChIP-qPCR assays for BRD4 and NF-κB binding at the aforementioned locus (black dot). Sequence enrichment was normalized to input. Data represented as mean ± SD. A two-sided, paired *t*-test was performed on percent input values (*n* = 8, biologically independent samples from three independent experiments). **d**, **e** ROSE algorithm of putative super-enhancer analysis from LSECs without (**d**) or with (**e**) TNFα treatment. The region contained in the orange dashed box contained sequences with top H3K27ac enrichment and are considered to be putative super-enhancers. **f** 3 C experiments were performed on LSECs to detect an interaction of the predicted CXCL super-enhancer with promoters of various CXCLs without (thick line) and with TNFα (thin line). The aforementioned NF-κB binding site (black dot) within the CXCL super-enhancer (dash lines) was used as a reference sequence. Interaction frequencies were plotted after normalized to that of *RASSF6*, a nearby noninflammatory gene used as control. Multiple other sequences were selected between target CXCL promoters as additional controls. X-axis maps relevant gene sequences as distance (in kb) from *RASSF6*. Two-way paired ANOVA analysis was performed on the relative interaction frequencies at target sites followed by post hoc Sidak's multiple comparison correction (*n* = 4, biologically independent samples from three independent experiments). Data represented as mean ± SD. Treatment with TNFα was found to be a significant variable (*p* = 0.0034). *CXCL1, 2, 3, 6,* and *8* loci were found to have increased interaction frequencies with *p* values as labeled.

epigenome editing (Fig. 4d). Interestingly, the expression of the dCas9-FLAG construct without the KRAB domain showed a similar pattern of repression, but to a lesser extent (Suppl. Fig. 17). This may be plausibly caused by steric hinderance to a crucial NF-κB binding site in the super-enhancer or by disruption of 3D chromatin looping related to the loss of CCCTC-binding factor (CTCF) binding, although no significant difference in chromatin interaction was noted by 3 C by dCas9-KRAB targeting (Suppl. Fig. 13). These experiments reflect the broad regulatory activity of CXCL super-enhancer on multiple CXCL genes, whereas repression of the *CXCL1* promoter has more specific effects exclusively on the *CXCL1* gene. We have also examined the occupancy of H3K27ac through ChIP-seq and

ChIP-qPCR in LSECs. H3K27ac occupancy in LSEC was increased with TNFα stimulation and decreased with Celastrol, which decreases NF-κB and BRD4 binding, suggesting that H3K27 acetylation may be dependent on NF-κB and BRD4 binding (Suppl. Figs. 11a, 12b).

To further validate the function of this super-enhancer in CXCL production, we proceeded to examine the impact of disrupting super-enhancer signaling pathway function using pharmacologic inhibitors. The transcription regulator and epigenetic reader BRD4 contributes to super-enhancer function by maintaining super-enhancer structure and facilitating the recruitment of other transcriptional cofactors. To reveal the role of BRD4 in LSEC super-enhancer function, we first studied

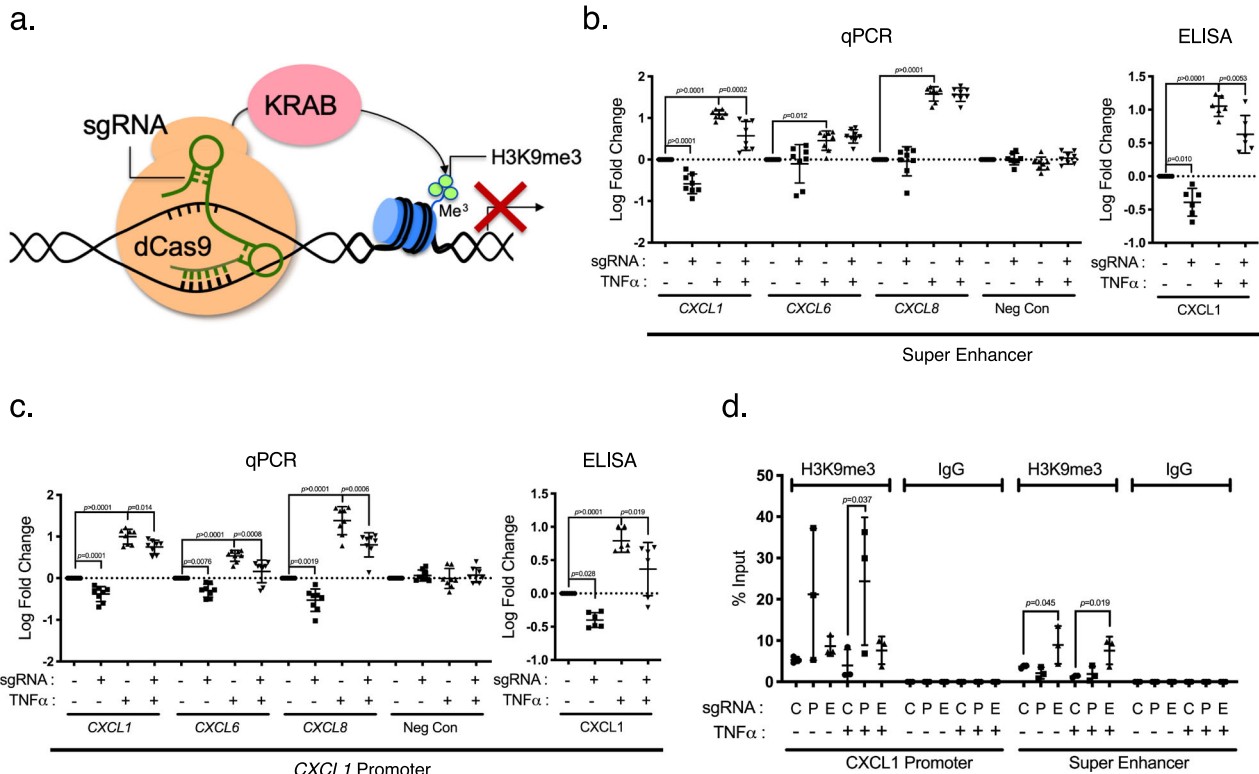

**Fig. 4 Histone modifications at CXCL super-enhancer and CXCL promoter sites modulate chemokine gene expression. a** Schematic of dCas9-KRAB binding with sgRNA leading to epigenetic silencing. **b** dCas9-KRAB fusion protein targeting the selected NFκB site within CXCL super-enhancer suppressed CXCL expression in LSECs. A sgRNA with specificity for a predicted NFκB binding site on the CXCL enhancer decreased *CXCL1, 6,* and *8* expressions revealed by qPCR but did not affect the expression of *MTHFD2L*, a nearby noninflammatory gene (negative control) ($n = 8$, biologically independent samples from four independent experiments). Changes in chemokine expression were calculated as fold change over basal expression and $\log_{10}$ (fold change) was plotted on the y-axis. ELISA for *CXCL1* was performed on supernatants and mirrored the pattern seen from qPCR ($n = 6$, biologically independent samples from three independent experiments). Data represented as mean ± SD. Two-way matched-pairs ANOVA was performed on log-transformed fold change values with post hoc Tukey's multiple comparison correction. **c** Another sgRNA targeting a predicted NF-κB binding site on *CXCL1* promoter decreased expression of *CXCL1,* but not other CXCLs by qPCR ($n = 8$, biologically independent samples from four independent experiments) and *CXCL1* ELISA ($n = 6$, biologically independent samples from three independent experiments). $\log_{10}$ (fold change) was plotted. Data represented as mean ± SD. Two-way matched-pairs ANOVA was performed on log-transformed fold change values, with post hoc Tukey's multiple comparison correction. **d** ChIP-qPCR for H3K9me3 on dCas9-KRAB treated cells. dCas9-KRAB was co-transduced with sgRNA targeting CXCL promoter (P), CXCL super-enhancer (E), or empty vector (C), and treated with or without TNFα. ChIP-qPCR was performed with anti-H3K9me3 antibody or isotype control. Enrichment for either *CXCL1* promoter or CXCL super-enhancer sequence was examined. Y-axis plots percent input ($n = 3$ from biologically independent samples). Data represented as mean ± SD. Two-way matched-pairs ANOVA was performed with post hoc Dunnett's multiple comparison correction.

iBET151, a commercially available pan-BET inhibitor that has highly specific activity against both bromodomains (BDs) of all four BET proteins (BRD2, 3, 4, and T)[33]. We found that LSEC CXCL expression was significantly decreased in the presence of iBET151 in vitro (Fig. 5a and Suppl. Fig. 18a) without significant cytotoxicity (Suppl. Fig. 19). Recent studies suggested that selective targeting of BD1 was sufficient for inhibition of BRD4 activity while potentially limiting clinical side-effects[34]. A novel BD1-selective inhibitor, UMN627, which has over a 20-fold affinity for BRD4 BD1 over BRD4 BD2[35], was tested on LSECs for its ability to suppress the activity of the CXCL super-enhancer. In vitro, UMN627 showed similar efficacy in CXCL suppression in stimulated LSECs as seen with iBET151 (Fig. 5b and Supp. Fig. 18b). We confirmed the inhibitory effect of iBET151 on BRD4 occupancy at both the CXCL super-enhancer and *CXCL1* promoter sites by ChIP (Fig. 5c). Interestingly, there was also a trend of decreased NF-κB binding at these sites after iBET151 treatment (Fig. 5d). A similar pattern was seen with Celastrol, which restricts NF-κB nuclear translation, leading to decreased binding of both NF-κB and BRD4 (Fig. 5e, f).

Considering the heterogenous sources of CXCL production in liver diseases (Suppl. Fig. 6), we sought to examine the cell-type specificity of this super-enhancer by analyzing ChIP-seq and Hi-C datasets from different cell types. The analysis of Hi-C data identified a significant overlap of topologically-associated domains (TADs) between HUVEC and LSECs at the CXCL locus, suggesting a similar local chromatin organization (Suppl. Fig. 20a)[36,37]. In a study with human lung fibroblasts, TNFα stimulation was found to enhance chromatin interactions between the CXCL super-enhancer and CXCL genes (Suppl. Fig. 20b)[38], indicating the activation of the CXCL super-enhancer. Another Hi-C study of human blood progenitor cells revealed similar enrichment of chromatin interactions in endothelial precursors, activated macrophages, and neutrophils (Suppl. Fig. 20c)[39]. However, there was no enriched interaction in anti-inflammatory macrophages, erythrocyte precursors, and naïve B cells, among others[39]. Indeed, the CXCL super-enhancer was ranked top in activated human macrophages in vitro by the ROSE algorithm[40]. Furthermore, there appears to be a homolog for this super-enhancer in mouse. For example,

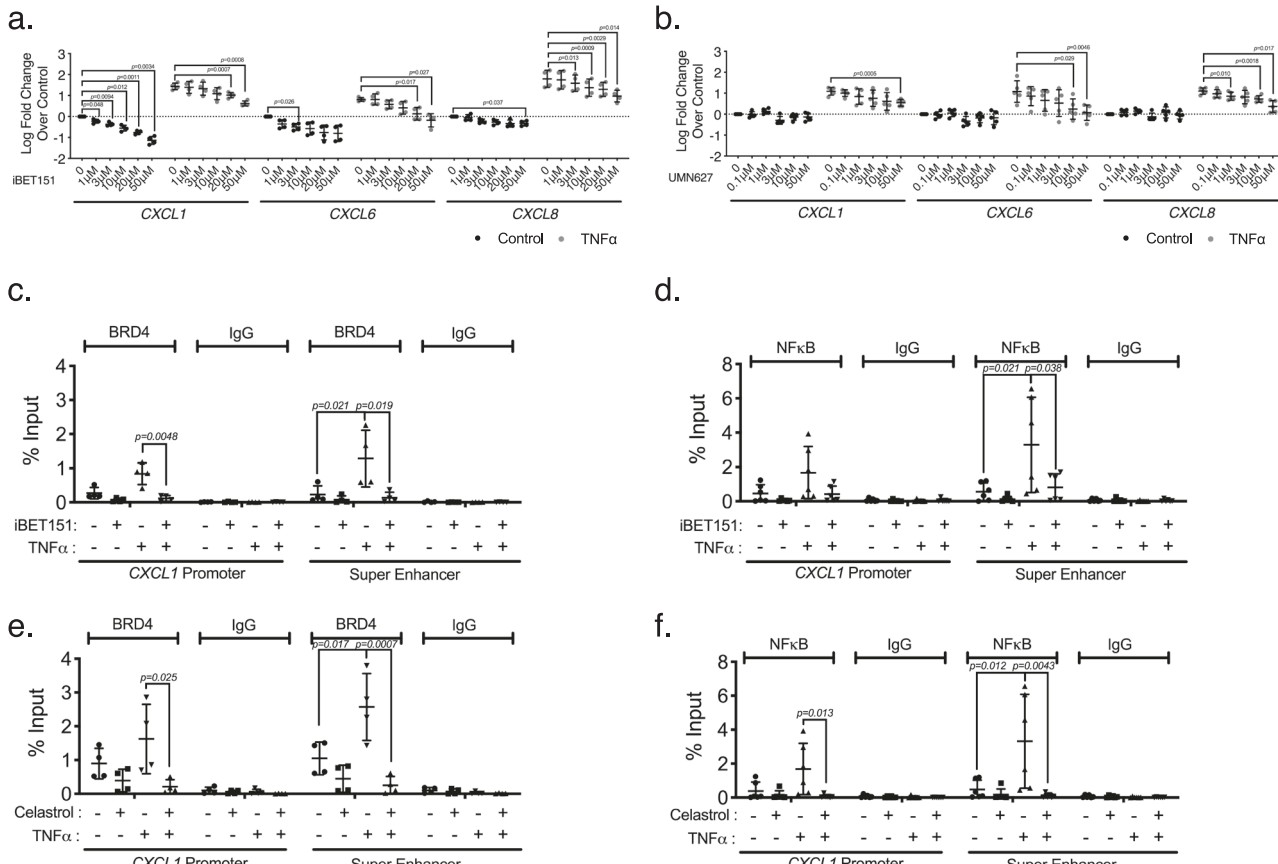

**Fig. 5 Bromodomain inhibitors suppress the expression of CXCLs by inhibition of transcription factor binding at CXCL super-enhancer and promoter sites. a, b** LSECs were pretreated with iBET151 (0–50 μM) ($n = 4$, biologically independent samples from two independent experiments) (**a**) or UMN627 (0–50 μM) ($n = 5$, biologically independent samples from three independent experiments) (**b**). *CXCL1, 6, and 8* expression levels were assessed by qPCR. Expression levels were normalized to basal conditions and log₁₀ fold change values were plotted. One-way matched-pairs ANOVA analysis was performed with post hoc Dunnett's multiple comparison correction. Data represented as mean ± SD. **c, d** ChIP-qPCR assays for BRD4 binding (**c**, $n = 4$, biologically independent samples from two independent experiments) or NF-κB binding (**d**, $n = 6$, biologically independent samples from three independent experiments) with and without TNFα stimulation was assessed at CXCL super-enhancer and *CXCL1* promoter sites after pretreatment with iBET151. One-way ANOVA analysis was performed with post hoc Tukey's multiple comparison correction. Data represented as mean ± SD. **e, f** Same experiments were repeated with Celastrol (**e**, $n = 4$ biologically independent samples from two independent experiments; **f**, $n = 6$ biologically independent samples from three independent experiments). Enrichment for either *CXCL1* promoter or CXCL super-enhancer sequence was examined. Sequence enrichment was normalized to input. One-way ANOVA analysis was performed with post hoc Tukey's multiple comparison correction. Data represented as mean ± SD. There were significant linear trends of decreasing *CXCL1, 6, 8* expressions with increasing iBET151 and UMN627 concentrations without and with TNFα, and with increasing iBET151 concentrations without TNFα ($p < 0.0001$ for all groups). All data were repeated at least three times with similar results.

Hah et al. identified two regions proximal to *Cxcl5 and Cxcl1* in the CXCL locus in mouse macrophages that were top-ranked as super-enhancers by ROSE[41]. ChIP-seq data also showed the enrichment of NF-κB and BRD4 occupancy within the two putative super-enhancer regions after LPS stimulation in bone marrow-derived macrophages in mice (Suppl. Fig. 21)[42,43]. Experiments with cultured macrophages support these observations as well. While isolated human circulating monocytes demonstrated poor response to TNFα stimulation, culturing monocytes with M-CSF induces differentiation into macrophages, and these cells become highly responsive to LPS and TNFα stimulation by upregulating chemokine expression (Suppl. Fig. 22). In a mouse primary hepatocyte RNA-seq study, stimulation with either TNFα or IL1β increased *Cxcl1* chemokine expression by 10- and 70-fold, under control and TNFα treatments respectively[44]. IL1β is another cytokine that is capable of activating the NF-κB pathway and was identified as a putative upstream regulator in our IPA analysis. In IL1β stimulated hepatocytes, ChIP-seq identified enriched occupancy of NF-κB and H3K27ac at the same two putative super-enhancer regions

(Suppl. Fig. 21)[44]. Taken together, these high-throughput epigenomic studies in both humans and mouse support a conserved role for the CXCL super-enhancer in regulating CXCL genes in immune cells and hepatocytes, as we have demonstrated in endothelial cells. Thus, we next proceeded to test the role of the CXCL super-enhancer in vivo.

**BET inhibition reduces Cxcl expression and neutrophilic infiltration in murine models of AH.** To test the biological relevance of our findings, we examined the effect of super-enhancer suppression in vivo. Using the NIH/NIAAA 10 days chronic-binge alcohol feeding protocol, we aimed to mimic the histopathology of human AH in mice[45]. In this model, *Cxcl1* gene expression was increased but not significantly and *Cxcl2* expression was not increased. *Cxcl1* and *Cxcl2* are functional homologs of human *CXCL8* and play important roles in neutrophil chemotaxis in mouse[46]. Immunohistochemistry (IHC) demonstrated modestly increased neutrophil infiltration with alcohol feeding (Suppl. Fig. 23). As expected, there was increased steatosis in

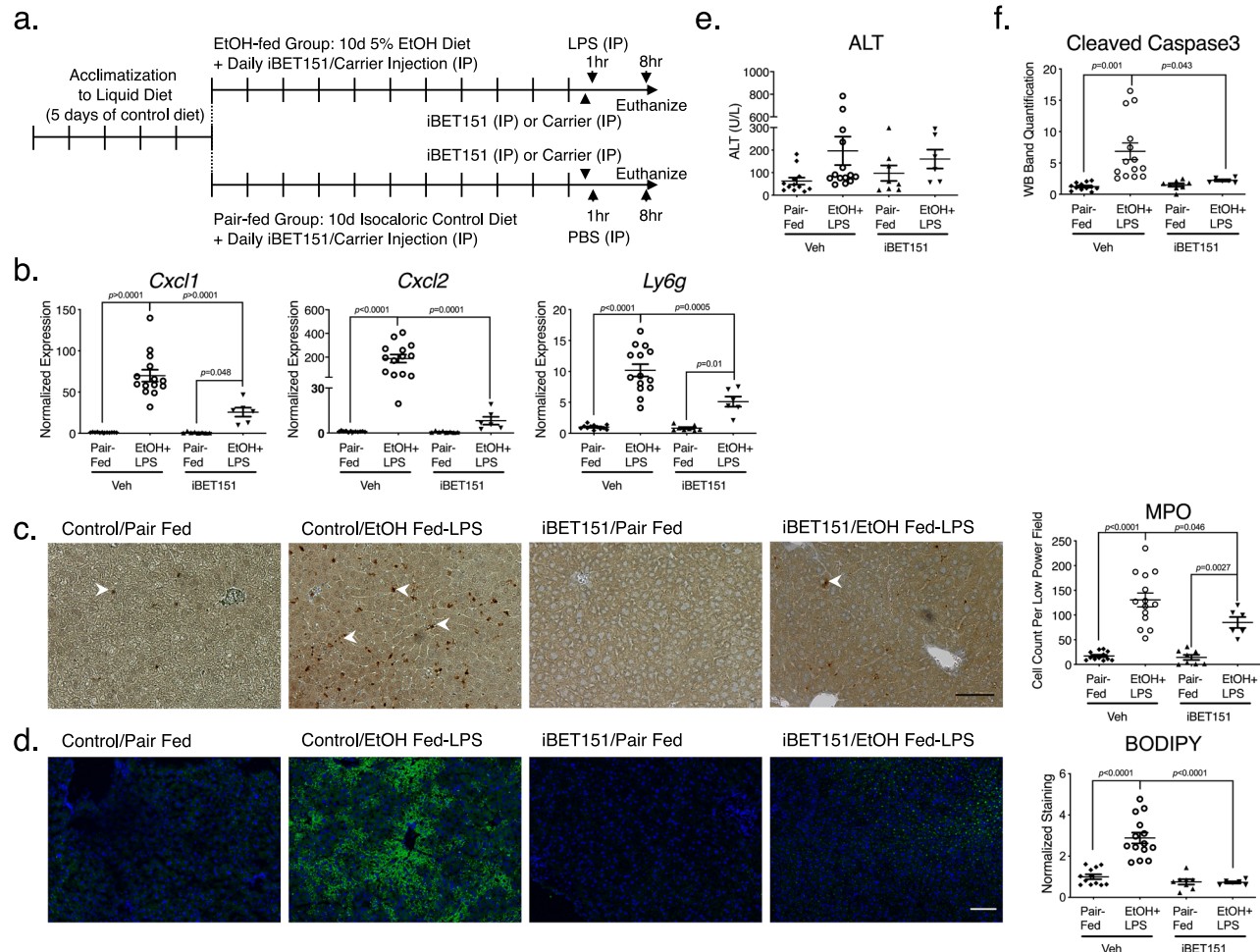

**Fig. 6 Bromodomain inhibitor iBET151 attenuates liver CXCL production and neutrophil infiltration in a murine alcohol feeding/LPS model. a** Schematic of alcohol feeding protocol. Mice were fed a 5% alcohol diet or a pair-fed control diet for 10 days. A subset of mice also received IP injection of the iBET151 compound. Alcohol-fed mice also received IP injection of LPS on day 11, 8 h before sacrifice ($n = 12$ pair-fed/Veh, $n = 14$ EtOH+LPS/Veh, $n = 8$ Pair-fed/iBET151, $n = 6$ EtOH+LPS/iBET151, same $n$ for remaining panels of this figure). **b** qPCRs demonstrated CXCL chemokine and neutrophil marker Ly6g elevation with alcohol/LPS treatment. This response was attenuated by iBET151 injections. Expression levels were normalized to the average expression of pair-fed control mice. **c** IHC for MPO (neutrophil marker) is shown, with a number of neutrophils per low power field on the y-axis. The experiment was repeated with similar results. Scale bar = 100 μm. **d** Frozen section of mouse liver was stained with BODIPY 493/503 (green) and DAPI was used to stain nuclei (blue). The experiment was repeated with similar results. Scale bar = 100 μm. **e** No statistical difference was noted in serum ALT levels among groups. **f** Quantification of Western Blotting of cleaved caspase3 showed increased cleaved caspase 3 in livers of alcohol-fed/LPS mice and attenuated by iBET151 injection. For all analysis, two-way ANOVA was performed on normalized expression values for qPCRs, cell counts for IHC staining, normalized quantification of BODIPY 493/503 staining, ALT values, or quantification of WB bands with post hoc Tukey's multiple comparison correction. Data represented as mean ± SD.

alcohol-fed mice as assessed by Oil-Red-O staining (Suppl. Fig. 23d). To generate an in vivo model with greater inflammation that could simulate human AH, we utilized a modified protocol of the chronic alcohol feeding model by substituting the alcohol binge with a single LPS injection as previously published by Kong et al. (Fig. 6a)[47]. Kong et al. demonstrated that chronic alcohol feeding and LPS injection synergistically induced liver inflammation and neutrophil infiltration. We similarly observed dramatically elevated hepatic expression of *Cxcl1* and *Cxcl2* in mice that underwent chronic alcohol feeding and LPS injection (Fig. 6b). Compared to mice given LPS injection alone, alcohol-fed/LPS mice showed increased *Cxcl1* and *Cxcl2* expression and worsened steatosis (Suppl. Fig. 24). In alcohol-fed/LPS mice compared to pair-fed mice, there was increased neutrophil infiltration of the liver demonstrated by higher *Ly6g* expression (Fig. 6b) and increased IHC staining for MPO (Fig. 6c). Increased steatosis by BODIPY stain (Fig. 6d) was observed in alcohol-fed/

LPS mice, but not in mice given LPS injection alone. To assess the role of super-enhancer activation in alcohol-fed/LPS-induced liver inflammation, the pan-BET inhibitor iBET151 was given to mice as daily intraperitoneal injections alongside alcohol feeding or LPS injection. Expression of *Cxcl1* and *Cxcl2* was significantly decreased in mice treated with iBET151 (Fig. 6b). iBET151 administration concurrently decreased neutrophil infiltration and steatosis (Fig. 6b, c). There was also a decrease in cleaved caspase 3 level in alcohol-fed/LPS mice given iBET151, suggesting suppression of apoptosis pathway activation with iBET151 administration (Fig. 6f and Supp. Fig. 25a). No difference was seen in cleaved caspase 3 levels between alcohol-fed/LPS mice and LPS only mice (Suppl. Fig. 25a). Lipid peroxidation marker malondialdehyde (MDA) and DNA oxidation marker 8-hydroxy-2-deoxy Guanosine (8-OHdG) were similar among treatment groups (Suppl. Fig. 25b, c). There was no statistically significant change in ALT levels (Fig. 6e). Our in vivo data suggest that

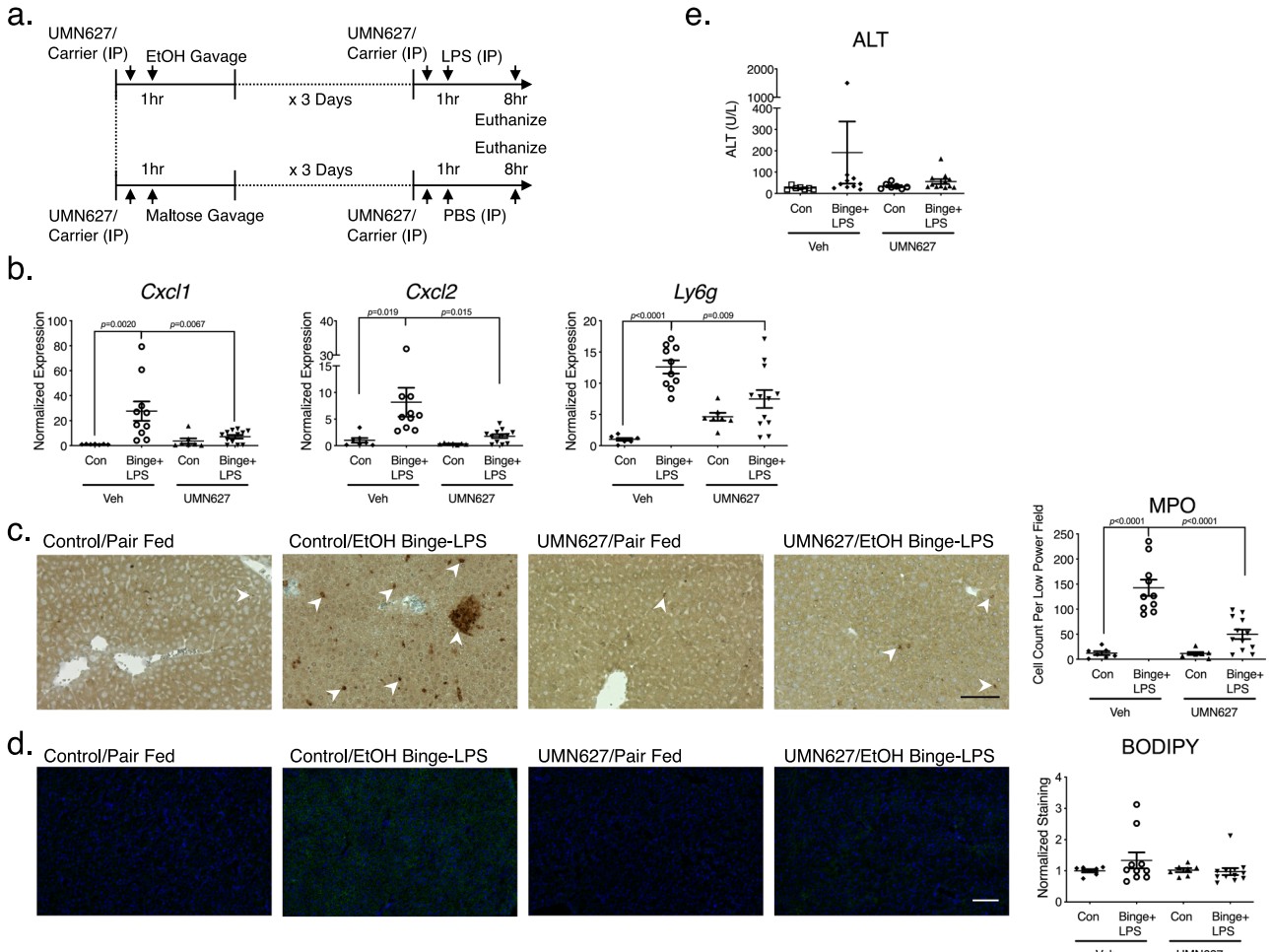

**Fig. 7 Bromodomain inhibitor UMN627 attenuates liver CXCL production and neutrophil infiltration in an alcohol binge/LPS model. a** Schematic of alcohol binge/LPS model protocol. Mice underwent once-daily gastric gavage with alcohol or maltose dextran for 3 days. A subset of mice also received IP injection of UMN627. Alcohol-gavaged mice also received IP injection of LPS on day 4, 8 h before sacrifice ($n = 7$ Control/Veh, $n = 10$ Binge + LPS/Veh, $n = 7$ Control/UMN627, $n = 12$ Binge+LPS/UMN627, same $n$ for remaining panels of this figure). **b** qPCRs demonstrated CXCL chemokine and neutrophil marker *Ly6g* expression elevation with alcohol/LPS treatment. This response was attenuated by UMN627. Expression levels were normalized to the average expression of maltose gavaged control mice. **c** IHC for MPO is shown, with a number of neutrophils per low power field on the y-axis. The experiment was repeated with similar results. **d** Frozen section of mouse liver was stained with BODIPY 493/503 (green) and DAPI was used to stain nuclei (blue). The experiment was repeated with similar results. **e** Serum ALT levels did not show statistical differences among various groups. For all analyses, two-way ANOVA was performed on normalized expression values for qPCRs, cell counts for IHC staining, normalized quantification of BODIPY 493/503 staining, or ALT values, with post hoc Tukey's multiple comparison correction. Data represented as mean ± SD.

suppression of super-enhancer activity by pharmacologic BET inhibitor decreases hepatic expression of *Cxcl* chemokines, which reduces liver inflammation in a murine model of human AH, consistent with our findings in LSECs in vitro.

In addition to the 10 days chronic alcohol feeding model, we have also experimented with a multiple alcohol binges/LPS injection model first published by Beier et al., who demonstrated that ethanol gavages accentuated liver injury caused by LPS injection[48] (Fig. 7a). Using this model, we were able to induce robust liver inflammation with increased CXCL expression and neutrophilic infiltration in alcohol binges/LPS mice (Fig. 7b, c). To ascertain if BD1-specific inhibition was sufficient in attenuating liver inflammation in this AH disease model, we administered UMN627 to mice undergoing alcohol binges/LPS injection[35]. We observed attenuated CXCL expression and neutrophil infiltration with the administration of UMN627 (Fig. 7b, c). There were no significant changes in ALT levels or steatosis among various treatment groups (Fig. 7d, e). Here, we explored another method

of inducing alcohol injury in mice and demonstrated that BD1-selective inhibition adequately attenuates alcohol/LPS-induced liver inflammation.

## Discussion

An estimated 35% of individuals with alcohol use disorder would develop AH, which when severe, is characterized by intense inflammation and liver failure[49]. In this study, we studied the transcriptional regulation of inflammatory signaling in AH and aimed to delineate the process by which hepatic inflammatory signals orchestrate the infiltration of immune cells into the liver parenchyma. We have made several observations in this study: (1) there were dramatic changes in the transcriptome and epigenome of human AH livers with several CXCL chemokines significantly upregulated; (2) the sources of these CXCL chemokine productions include liver sinusoidal endothelial cells and infiltrating immune cells in humans under the regulation of TNFα/NF-κB

signaling axis; (3) there is a super-enhancer to CXCL chemokines located upstream of the CXCL locus in liver cells; (4) suppression of this super-enhancer by CRISPR dCas9-KRAB or by BD inhibitors (either pan-BET or preferred binding to BRD4 BD1) repressed CXCL expression; and (5) BD inhibitors reduced liver neutrophilic infiltration by suppressing CXCL expression in murine models of AH.

Here, we identified a family of CXCL chemokines, including CXCL1, 6, and 8, that were highly upregulated in the livers of patients with AH. Previous publications have made similar observations, and increased expression of CXCL6 and CXCL8 have been correlated with worsened clinical outcomes in AH[16,50]. CXCL chemokines, as potent chemoattractants to neutrophils, provide a direct molecular mechanism for the neutrophilic infiltration seen in AH. Although the importance of CXCL chemokines in AH has been recognized previously, the source(s) of these CXCL chemokines in the liver have remained largely unknown. Multiple cell sources have been reported to be responsible for the production of CXCL chemokines in AH, including hepatocytes, stellate cells, Kupffer cells, ductular reactive cells, endothelial cells, and neutrophils themselves[22,24,28,51,52]. In this study, we analyzed transcriptomic data of normal human liver cells, FANTOM5 transcriptome data, and published scRNA-seq data in humans and mouse to identify LSEC as an important producer of CXCLs in the liver (Suppl. Table 1). Endothelial cells affect broader organ pathophysiologic changes by releasing soluble factors, such as chemokines, in an increasingly recognized process termed "angiocrine signaling"[53]. Murine LSEC has been demonstrated to upregulate Cxcl1 and other chemokines in response to TNFα given exogenously or secreted by Kupffer cells, resulting in immune cell infiltration[51,54]. As resident liver cells with direct exposure to intravascular signals, LSEC is uniquely positioned to not only sense but also participate in the liver inflammatory response. We propose that LSEC are among the "first responders" to inflammatory cascade activation, and by recruitment of infiltrating immune cells to the liver parenchyma, set off a positive feedback loop that quickly amplifies the inflammatory signal[28,53]. In AH, LSECs and immune cells are important sources of chemokine production; although technically challenging, further investigations with scRNA-seq of human AH liver tissues could potentially elucidate the relative contributions of resident liver cells vs. infiltrating immune cells in AH pathogenesis. Among multiple CXCL chemokines involved in neutrophil chemotaxis, only CXCL2 was found to be downregulated in AH. Prior publications have observed similar findings[50]. In silico analysis of FANTOM5 database and scRNA-seq data[55] suggested that CXCL2 is most highly expressed in hepatocytes in humans (Suppl. Fig. 6 and Suppl. Table 1), and hepatocyte injury found in AH may decrease expression of CXCL2, either through transcriptional downregulation or via a reduction in hepatocyte number. The functional role, if any, of the differential expression of CXCL2 and other CXCL chemokines is unknown, which may also be more definitively investigated with future single-cell sequencing studies in AH tissue or disease models.

Murine Cxcl chemokines are homologous with human counterparts but may differ in their cellular origins. For example, Cxcl1 may be most abundantly produced by hepatocytes in normal mouse liver, whereas human hepatocytes were not shown to be an important source of CXCL production in our study and other bulk RNA-seq or scRNA-seq studies (Suppl. Fig. 26 and Suppl. Table 1)[56]. Despite the difference in the cellular source of chemokine production, high-throughput sequencing studies suggest that CXCL regulation in many mouse cell types, including macrophages and hepatocytes, is likely similar to humans, which utilizes the CXCL super-enhancer. This inference is supported by our experimental data showing excellent anti-inflammatory

responses of BD pharmacological inhibition in mouse models[34]. It is important to note these differences in cellular origins as therapeutics targeting of CXCL production may need to tailor to species-specific sources, and agents that target multiple cell types, as we did here with BET inhibitors, may be best suited for cross-species investigation.

Given the remarkable increase in chemokine production in AH, we set out to elucidate the signaling pathways leading to this upregulation. TNFα is a well-studied inflammatory cytokine in the pathogenesis of AH[3] and a known stimulus for CXCL1 release in vascular endothelial cells[13]. In this study, we demonstrated that TNFα/NF-κB signaling robustly stimulated CXCL expression. Earlier work by Brown et al. has implicated a role of super-enhancers in regulating TNFα-induced genes in the context of inflammation[13]. Here, using 3D chromatin interaction assays, ChIP-seq in AH livers and LSECs, and public ChIP-seq datasets, we identified a super-enhancer in LSECs that controls the expression of multiple CXCL chemokines under the modulation of TNFα. This locus has recently been functionally verified as a super-enhancer for CXCLs in HUVEC cells[29,30] in addition to being identified in Hi-C studies to interact with CXCL1 promoter[57]. Here, we were able to identify the presence of this super-enhancer in LSECs and demonstrate its functional importance by epigenetically targeting an NF-κB binding site in this region with dCas9-KRAB. Identification of this CXCL super-enhancer in multiple cell types along with the plurality of cellular sources of CXCL chemokine production suggests that this mechanism of chemokine regulation is shared across different cell types. Therefore, interventions focused on disruption of the super-enhancer function could be beneficial through effects on multiple cell types. We also highlighted the role of histone modifications in gene regulation in AH. There is H3K27ac enrichment on the promoter and super-enhancer of CXCL chemokines in response to TNFα stimulation in LSECs and in AH livers. We further linked the enrichment of H3K27ac with transcription factor binding, and pharmacologic inhibition of NF-κB and BRD4 binding attenuated TNFα-induced H3K27ac enrichment as well as decreased CXCL expression. NF-κB is known to mediate p300 recruitment to target sites[58], and epigenetic histone modification may help to amplify TNFα signaling and propagate the inflammatory response. In addition, the upregulation of epigenetic modifier pathways with SP1 and SMARCA4 further supports the role of epigenetic control in AH gene regulation. These findings, together with our ability to epigenetically target the super-enhancer for transcription regulation, highlight the importance of epigenetic regulation in AH and may provide novel strategies to modulate the CXCL super-enhancer function.

Over the last decade, many pharmacologic agents targeting various transcriptional mediators in the canonical super-enhancer signal transduction pathways have emerged for the treatment of various malignancies[59]. BET proteins are transcriptional mediators that play essential roles in the maintenance of chromatin architecture and regulation of super-enhancer activity. Of these BET proteins, BRD4 appears to be the main protein responsible for maintaining gene expression programs[34,60]. BRD4 inhibition is particularly appealing for super-enhancer interference not only because of its importance in super-enhancer function but also because enhancer-bound BRD4 may be more susceptible to pharmacologic inhibition than promoter-bound BRD4, thus enhancing the specificity of treatment[61]. However, clinical application of inhibitors to BRD4, particularly pan-BET inhibitors, has been limited by pharmacological side-effects[62]. To improve target specificity, novel agents have recently been reported that target only BD1 domain of BET proteins, including BRD4, and BD1 inhibition alone has been demonstrated recently to be sufficient for BRD4 inhibition[34,35,63] and effective in attenuating airway

inflammation[64]. Recent publications indicate that BET inhibitors only repress BRD4 activity at a subset of target genes, and BRD4 binding at unresponsive genes is not BD-specific and may in fact preclude access of BD-specific inhibitors[34,65]. Therefore, the use of BD-specific inhibitors may further improve therapeutic targeting to only inhibitor-responsive target genes. A novel BD inhibitor tested here, UMN627, has preferential binding to BRD4 BD1[35]. We observed that UMN627 suppressed CXCL production both in vitro and in a murine AH model. Compared to a pan-BET inhibitor, BD1 specific inhibitors have the potential benefit of fewer side-effects with similar BRD4 inhibition. To study the in vivo effect of these compounds, we utilized multiple animal models of human AH. The widely adapted NIAAA chronic-binge murine AH model induced a modest degree of liver inflammation and incompletely recapitulated the massive neutrophilic infiltration seen in severe human AH. Here, we employed two modified alcohol injury models, a chronic murine alcohol feeding model and a multiple alcohol binges model with the addition of a single LPS injection[47,48]. LPS is a bacterial product that has a well-recognized role in the pathogenesis of AH, particularly with regard to immune activation[3]. Chronic alcohol use leads to increased intestinal permeability and dysbiosis that causes elevated LPS levels in the enterohepatic circulation, a process aptly termed the "Gut-Liver Axis"[3]. Our upstream regulator analysis of AH upregulated genes also identified LPS stimulation as a top potential target (Fig. 1e). The addition of LPS has a synergistic effect with alcohol feeding to better simulate the profound immune activation seen in human AH. This model allowed us to show unequivocally that BD inhibitors decreased CXCL chemokine levels and attenuated liver neutrophilic infiltration in mice. Cleaved caspase 3 level is also decreased in BET inhibitor-treated mice, possibly due to decreased cell-mediated cytotoxicity that accompanies attenuation of immune cell infiltration (Fig. 6f). This result demonstrates that BRD4 is essential for the stimulation of CXCL chemokine production following activation of inflammatory signaling. This may be due to structural disintegration of the CXCL super-enhancer, but inhibition of BRD4 may have additional direct effects on NF-κB signaling[63]. For example, BRD4 has been shown to interact directly with activated NF-κB and help stabilize NF-κB within the nucleus[66]. This may help to explain the decreased NF-κB enrichment at the CXCL super-enhancer and promoter sites in the presence of BD inhibitors in LSEC cells. Collectively, we demonstrated that pharmacologic inhibitors of BET proteins suppressed CXCL super-enhancer-regulated chemokine production in mice, closely mirroring our observations in vitro. Interestingly, BET inhibitor iBET151 was also effective at attenuating alcohol feeding induced steatosis in the 2-week alcohol feeding model; this finding is consistent with an earlier report that iBET151 is effective at reducing steatosis in a NASH model[34,67]. The anti-steatosis effect of iBET151 may be due to its anti-inflammatory properties or direct effects on lipogenesis.

Therapeutics aimed at suppressing inflammatory responses in the liver has failed in the treatment of AH[68]. To date, corticosteroids remain the sole widely accepted therapy for severe AH despite having only modest clinical benefits. Direct TNFα inhibitors have been studied in clinical trials but failed to improve outcomes due to profound immunosuppression and impaired liver regeneration[68]. Given the diverse functions of TNFα in liver recovery, successful therapeutics for AH may require specific targeting of inflammatory pathways downstream of TNFα. Given their broad activity against a large number of inflammatory genes and specificity for their target genes, super-enhancers make attractive candidates for pharmacologic intervention. Here, we identified a super-enhancer intricately involved in the progression of AH, which provides a rationale for the use of pharmacologic inhibitors targeting super-enhancer function in the treatment of AH.

## Methods

**Human study participants**. For human liver RNA-seq and ChIP-seq analyses, human liver explants from six patients with AH were procured at the time of liver transplant surgery. Four control patients with no history of chronic liver diseases undergoing liver resection for other causes were obtained to serve as controls. Select demographical information on these patients was provided in Suppl. Fig. 1.

**RNA-seq and histone mark ChIP-seq**. Human liver tissues were processed at the Mayo Clinic Center for Individualized Medicine Medical Genomics Facility for RNA-seq and Epigenomics Development Laboratory for ChIP-seq on H3K27ac, H3K4me1, H3K4me3, and H3K27me3 marks. Liver cell RNA-seq was performed at the Center for Individualized Medicine Medical Genomics Facility as well. RNA-seq utilized samples from all six patients, and ChIP-seq was performed on five of six patients. Further details were provided in supplemental methods.

**Cell culture and TNFα stimulation**. Primary human LSECs were purchased from ScienCell (Cat #5000) and cultured using standard cell culture techniques. For the liver cell RNA-seq experiment, primary human LSECs, HSCs (Cat #5300), and HiBECs (Cat #5100) were purchased from ScienCell, and the HepG2 cell line was obtained from ATCC (HB-8065). Lentiviral transduction experiments of LSECs were described in detail in supplemental methods. Where appropriate, LSECs underwent TNFα (Peprotech, 300–01 A) stimulation at 20 ng/mL. Detailed procedure of TNFα stimulation was described in Supplemental Methods. In selected experiments, cell supernatant was collected and enzyme-linked immunosorbent assay (ELISA) was performed to assess the concentration of secreted CXCL1.

**Real-time PCR**. mRNA levels were quantified by real-time reverse transcription PCR. Further details are provided in supplemental methods. Primers used for qPCR were listed in Suppl. Table 4.

**4 C**. Brief experimental procedure was described in supplemental methods.

**3 C**. Brief experimental procedure was described in supplemental methods.

**ChIP**. LSECs were treated with appropriate conditions as outlined separately and underwent ChIP according to Millipore High-Sens ChIP kit (Millipore MAGNA0025) manufacture protocols. Briefly, cells were crosslinked with formaldehyde (1% final concentration) followed by glycine treatment (100 mM) for 5 min each. Cells were washed, collected, pelleted with centrifugation, and lysed with cell lysis buffer. Cells were repelleted and underwent nuclear lysis with provided nuclear lysis buffer, and DNA was sheared with ultrasonication. Soluble chromatin was aliquoted and immunoprecipitated with magnetic beads with antibodies for BRD4 (Abcam ab128874), NF-κB (Cell signaling 8242 S), H3K9me3 (Abcam ab8898), or H3K27ac (Abcam 4729) with appropriate isotype controls, 5 μg per immunoprecipitation reaction. Immunoprecipitated beads were collected and processed according to the manufacture's protocol. Real-time PCR was performed in purified ChIP and input DNAs at target loci, and enrichment was compared with isotype control IgG.

**In vitro pharmacologic inhibitor assays**. LSECs were treated with Celastrol (Sigma C0869), iBET151 (Cayman 11181), or UMN627 (provided by Dr. Pomerantz from the University of Minnesota). Cells were harvested and gene expression was assessed by real-time PCR. Further experimental details were provided in supplemental methods.

**Immunohistochemistry**. See Supplemental Methods.

**Animal experiments**. All animal experiments were performed in accordance with the regulations of the Mayo Clinic Institutional Animal Care and Use Committee in AAALAC-accredited facilities. Protocols were reviewed and approved by the Mayo Clinic Institutional Animal Care and Use Committee.

**Chronic-binge alcohol feeding model**. WT C57BL/6 mice (10–12 weeks) were purchased from Envigo Laboratories. Mice were subjected to chronic-binge alcohol feeding to induce alcohol-induced liver injury in accordance with the NIAAA model[45]. Briefly, mice were acclimated to a liquid diet for 5 days and were fed either a 5% alcohol-containing diet (Bio-Serv F1258SP) or an isocaloric pair-fed control diet (Bio-Serv F1259SP). All mice were gavaged with ethanol 6 g/kg (alcohol-fed mice) or an equal volume of maltose dextran (pair-fed mice) on day 11, and all mice were sacrificed 9 h later.

**Alcohol feeding with LPS model with BRD4 inhibitor iBET151**. WT C57BL/6 mice (10–12 weeks) were purchased from Envigo Laboratories. Mice were subjected to chronic alcohol feeding to induce alcohol-induced liver injury with modification to the NIAAA model[47]. Briefly, mice were acclimated to a liquid diet for 5 days and were fed either a 5% alcohol-containing liquid diet or an isocaloric pair-fed control

diet. On day 11, IP injection of LPS (Invivogen tlrl-eblps) at 4 mg/kg was administered in alcohol-fed mice whereas PBS of the same volume was given to pair-fed mice. In LPS injection only mice, animals were fed a control diet and given LPS injection on day 11. All mice were sacrificed 8 h after LPS injection. In a subset of mice, concurrent with alcohol or pair-feeding, they were given an injection of iBET151 daily. Mice were injected intraperitoneally with iBET151 (6 mg/kg) in 10% Kleptose 2% DMSO solution or an equal volume of carrier solution. On day 11, the drug or vehicle was administered 1 h before LPS injection.

**Multiple alcohol binge with LPS model with BRD4 inhibitor UMN627.** WT C57BL/6 mice (10–12 weeks) were used in this model. All mice were fed standard chow and underwent gavage with 6 g/kg alcohol solution or equal caloric maltose dextran solution once a day for 3 days. On day 4, mice receiving alcohol gavages were given IP injection of LPS at 4 mg/kg and control mice were injected with an equal volume of PBS, and all mice were sacrificed 8 h later[48]. UMN627 (22 mg/kg) in 10% Kleptose and 1% DMSO or an equal volume of carrier solution were administered by IP injection to mice 1 h before gavage each day or LPS/PBS injection on day 4.

**Statistical analysis.** Mean is expressed as mean ± standard deviation. Statistical analysis was conducted using GraphPad PRISM (La Jolla, USA) and R statistical software. Comparisons between three groups or more were conducted using one-way ANOVA with Dunnet's or Tukey's post-test for multiple comparisons using GraphPad PRISM. Comparisons with two different conditions were performed with two-way ANOVA with Sidak's or Tukey's post-test for multiple comparisons. A comparison of two groups was performed using the Student's $t$-test. $P$ value $\leq 0.05$ is considered significant.

**Study approval.** Human tissues used for RNA-seq and ChIP-seq analysis were collected at the University of Lille, France under institutional IRB protocol, deidentified, and sent to Mayo Clinic for further analysis. Normal human tissues and primary human cells were collected at Mayo Clinic under institutional IRB approval. The aforementioned IRB protocols were approved by the Ethics Committee of the University of Lille, France, and Mayo Clinic, respectively. Written, informed consent was obtained from all study participants. The study was conducted in accordance with the Declaration of Helsinki. The use of primary human cells was approved by the Mayo Clinic Institutional Review Board.

**Reporting Summary.** Further information on research design is available in the Nature Research Reporting Summary linked to this article.

## Data availability

All RNA-seq and ChIP-seq data generated in this publication are available on the GEO database (GSE155926 https://www.ncbi.nlm.nih.gov/geo/query/acc.cgi?acc=GSE155907 and 166564). JASPAR database is accessible via weblink http://jaspar.genereg.net/. All relevant data generated during this study are included in this published article, supplementary information files, and Source Data files. Source data are provided with this paper.

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

## Acknowledgements

This work is supported by funding provided by the National Institutes of Health (NIH), USA R01 AA21171 and R01 DK59615 (VHS).

## Author contributions

The research was designed by S.C., V.H.S., and M.L. Human tissues were procured by P.M. and R.B. J.P.A., S.C., J.-H.L., and H.Y. performed the genomic sequencing studies and associated data analysis, with additional analytic studies by M.L., S.C., F.H.H., and S.A.J. Microfluidic studies were performed by Y.G., M.L., T.S.S., and A.R. Western blot and ChIP experiments were performed by L.H. Experimental reagents and microfluidic devices were provided by M.V.-C. and A.R. Chromosomal conformation capture assays were performed by S.C., M.L., and H.Y. CRISPR directed cell-based experiments were performed by M.L. and J.G. with input and reagents provided by F.G. and T.O. UMN627 was synthesized and provided by H.C. and W.C.K.P. Animal experiments were performed by M.L., H.L., and T.S.S. WB and cytotoxicity assay was performed by M.L., H.L., and W.X. Statistical analysis was performed by M.L. Manuscript was prepared by M.L., S.C., V.H.S., S.A.J., T.O., and H.Y.

## Competing interests

The authors declare no competing interests.
