## [Peer Review File · Nature Communications]

Reviewers' Comments:

Reviewer #1:

Remarks to the Author:

In this paper Liu et al study the mechanistic role of super enhancers in chemokine expression in the context of alcoholic hepatitis. Combining studies in human liver samples, in vitro studies and in vivo studies in mice exposed to bromodomain inhibitors, they show that epigenomic changes in livers from patients with alcoholic hepatitis is associated with up regulation of CXCL chemokines. They identify super enhancer to CXCL chemokines upstream of the CXCL locus, in particular in liver endothelial cells, that govern CXCL expression, as shown when silencing this super enhancer by CRISPR dCas9-KRAB or by pharmacological bromodomain inhibitors. Finally, using murine models of alcoholic liver disease with more or less severity, they demonstrate that BET inhibitors reduce liver expression of CXCL and infiltration of neutrophils. The study, performed by a well established group in the field, is interesting and novel, and contains an impressive amount of data. However, I have several major concerns, in particular regarding in vivo studies, that need to be addressed and found some data overinterpreted.

1. The reason as to why the study on isolated cells was exclusively focused on endothelial cells is not clear and relies on data from normal liver excluding immune cells. The fact that LSEC is the predominant cell type expressing CXCL chemokines in the normal liver is not a sufficient argument to exclude immune cells which are a major source of inflammatory mediators, most likely including CXCL in AH, as underlined in Sup Fig 13. This is of importance since in in vivo studies, pharmacological inhibitors will not only target endothelial cells.
2. In in vivo studies, reduction of CXCL expression is shown in two models of alcohol feeding. Nevertheless, a major issue in the context of the disease is to reduce inflammation but also in liver injury. However, the authors limit their studies to chemokine levels quantification and MPO. Strikingly, they do not investigate steatosis (which surprisingly when looking at IHC is not obvious), and provide results on ALT that are either not convincing or come from a very low number of animals. Caspase and oxidative stress should also be assessed
3. In in vivo studies, the number of samples analysed vary with the parameter studied. For example in Fig 6, MPO results are n=4 for ETOH LPS, but n=16 for Ly6G, or n=14 for CXCL. In sup Fig 14 outliers were excluded for CXCL, and ALT data are analysed in n=4 for ETOH and more for CXCL etc...6-8 pair fed are indicated in the legend and 12-14 alcohol?. Why do the authors do not show the results of all treated animals and the number shown vary with parameters? Same comments for Fig 7, for example for ALT
4. The main effect of sg RNA in Fig 4B and C is to reduce basal expression of CXCL. Viability tests showing the lack of toxicity should be provided. Same comments for bromodomain inhibitors in Fig 5 .

Additional comments.

Which CXCL is assessed by ELISA in Fig 2B?

In exp with LSEC medium provided in sup Fig 8, it looks like if LSEC are dying ?

In Fig 5A for each chemokine, data shown are - and + TNF?

Reviewer #2:

Remarks to the Author:

Comments for the authors

The studies in this paper use a variety of sophisticated methods to examine the changes in CXCL expression and underlying regulatory mechanisms in liver samples from patients with alcoholic hepatitis and the potential therapeutic effectiveness of bromodomain inhibitors for alcoholic hepatitis in two mouse models. The major advances of the study are: 1) the identification of sinusoidal endothelial cells as a major contributor of CXCL expression; 2) sophisticated confirmation of the long range interactions of an enhancer region regulating multiple CXCL genes upstream of the genes; and 3) the demonstration of the effectiveness of pan-bromodomain inhibition or specific BD1 inhibition in attenuating alcoholic hepatitis in mouse models. However, there are significant concerns. Evidence for the specific role of alcohol in the regulatory mechanism

proposed is not presented, so it is not clear whether alcoholic hepatitis or general liver inflammation and TNF α /NF- κ B signaling is being studied. The epigenetic studies are weak and not complete and evidence for epigenetic control and a "super" enhancer is weak. The roles of a super enhancer regulating CXCL expression in mouse and humans and CXCL involvement in alcoholic hepatitis are not novel.

1. The studies with the KRAB construction to demonstrate epigenetic control is not compelling. The control dCas9 without KRAB has a similar, but smaller effect suggesting that another mechanism is involved, perhaps steric hindrance as mentioned, which might be enhanced by the additions of the KRAB to the dCas9. Evidence for epigenetic control is therefore weak.

2. If the emphasis of the study is on epigenetic studies, then, the important epigenetic modifiers that are altered in alcoholic hepatitis should be identified.

3. Relative to the KRAB studies, if this is a super enhancer with multiple NF- κ B sites, why would targeting a single NF- κ B site have such a dramatic effect, i.e. reducing the effectiveness of the entire complex of enhancers. Was there a change in the epigenomic landscape of the entire super enhancer by targeting one NF- κ B site?

4. As noted by the authors, a super enhancer regulating CXCL expression in mice and humans and a role of CXCL in alcoholic hepatitis have been previously described, which reduces novelty of the findings in this paper.

5. What is being studied in the mouse models is confounded by the stimulation with LPS. It is not clear whether the changes in regulation of Cxcl are due to alcohol toxicity or to general inflammation with LPS. Comparison of alcohol + LPS with LPS might allow changes specific to alcoholic hepatitis to be identified.

6. Data presentation could be improved.

- Box and whisker plots are useful for larger datasets, but n values in most of these experiments are small and all points should be plotted as in Fig. 6 and 7.

- Also, using log plots are useful if there are order of magnitude differences, but linear plots are more intuitive and better for less than 100-fold differences, which is the case for most of these plots that use the log scale.

- Fig. 4D, plots use derived data of fold difference and normalization to control for a ChIP-seq. Better to plot more direct data as % input as in Fig. 3C and 5C-F.

Minor

L61 – delete "on"

L182 – prorogating = propagating

L217 – chemokine – chemokines

L231-234 – BRD4 is analyzed in Supp. Fig. 9B and should be mentioned here.

Fig. 5C-F – indicating the antibody of control IgG used with + and – below the graph can be confusing suggesting for example that BRD4 in C is expressed in the cells. Better to indicate BRD4, etc. and IgG above the bar plots.

L322, 325, 327 – macrophage = macrophages

L331 – expressions = expression and folds = fold

L343 – In Supp. Fig. 14, Cxcl2 expression did not increase contrary to the statement in the text. Also Ly6g which is measured in this fig should be mentioned

L355 – BODIFY staining for steatosis in Supp. Fig. 15 is increased as described, but there is no mention of Oil Red O staining in Supp. Fig. 14.

L361 – ALT levels and Steatosis did not show statistically significant differences with iBET151 as stated, but this is a little misleading, suggesting no difference. The data show a decrease trend that was not significant (Fig. 6D and Supp. Fig. 15)

L448 – additional = addition

L487 – simulate = simulates

Reviewer #3:

Remarks to the Author:

The manuscript by Liu and colleagues describes a study that identifies mechanisms regulating chemokine production from the Cxcl locus in human liver from patients suffering from alcoholic hepatitis. The authors first use a genomics approach (a combination of RNA-seq and ChIP-seq against various histone modifications) to find differentially regulated genes and enhancers in human liver biopsies from healthy and diseased donors. Among TNF α and NF- κ B controlled

inflammatory pathways, they find differential regulation of Cxcl1,6 and 8 expression. Using a combination of publicly available data and gene expression analysis in different hepatic cell types they argue that Cxcl1,6 and 8 are predominantly expressed in human liver sinusoid endothelial cells (LSECs). Based on genome wide H3K27Ac in human umbilical vein endothelial cells (HUVECs) and chromatin conformation assays (3C and 4C-seq) in LSECs they suggest that Cxcl1,6 and 8 expression is controlled by a super enhancer in close vicinity to Cxcl8. The enhancer is occupied with NF-kB and BRD4 in response to TNFalpha in LSECs in line with differential Cxcl1,6 and 8 expression upon an inflammatory response. Using Cas9-KRAB and gRNA against the putative super enhancer they convincingly demonstrate that the enhancer is needed for maximal expression of Cxcl1,6 and 8. Lastly, they demonstrate that BED inhibitors can attenuate Cxcl1,6 and 8 expression and lower the inflammatory response in a mouse model of alcoholic hepatitis.

Identification of mechanisms controlling neutrophil recruitment to the inflamed liver is of high interest in order to develop targeted anti-inflammatory strategies in tissue such as the liver. This study suggests a novel mechanism for cytokine induced expression Cxcl1,6 and 8 and based on this suggests an anti-inflammatory strategy targeting regulatory enhancer activity by BED inhibitors. The study is overall well designed, and the individual experiments are sound. However, there are a couple of concerns relating specifically to the LSECs as the major contributor to Cxcl1,6 and 8 expression in the inflamed liver and the suggestion that expression is controlled by a so-called super-enhancer.

Major concerns:

The authors perform a fairly robust analysis of RNAseq and histone mod ChIPseq. However, details on sequencing depth and uniquely aligned reads/fragments for the individual libraries is not provided. This should at least be listed in a table so the reader can evaluate the NGS data quality.

Cell type specific expression of Cxcl's was first assessed by mining the FANTOM5 database and confirmed using RNAseq in different primary human liver cell types. Unclear from the material and methods how these primary cell types were isolated from human liver. Should be clarified in the material and methods. Also, to further validate the cell type specific expression of Cxcl's the authors should show expression of relevant Cxcl's from previously published human liver single cell data (PMID: 31292543, PMID: 30348985 and PMID: 31597160). This will specifically address which cell types of the liver that express the different Cxcl genes.

The illustration of Cxcl locus shown in supplementary figure 7 could be improved. Would be informative to show the locus with relative distance between the genes and enhancers. Here a browser shot from for example the UCSC genome browser would be helpful. Here H3K27Ac could also be shown to highlight the position of the super enhancer. Also, the indicated predicted NF-kB binding sites should be shown.

Experiments using TNFa and a NF-kB inhibitor treatment of primary LSECs shown in figure 2B and C suggest that NF-kB is involved in Cxcl1,6 and 8 expression in LSECs. This may be directed by NF-kB binding to the promoters and the super enhancer indicated in Suppl. Figure 7. Would be informative to show NF-kB recruitment to the predicted NF kB binding sites at promoters and super enhancer using ChIP. Although the NF-kB ChIP is shown for one site (super-enhancer) in figure 3C, it would be informative to probe NF-KB at all suggested binding sites. Also is H3K27Ac at the promoters, enhancers and super enhancer changed in response to TNFa treatment and inhibited by Celastrol? This would indicate if promoter and enhancer activity is regulated by TNFa and NF-kB.

The Cxcl2,3 and 5 are in vicinity of the Cxcl1,6 and 8 genes. Are Cxcl2, 3 and 5 expression changed by TNFa treatment in LSECs? Also are Cxcl2, 3 and 5 differentially regulated by AH in liver? Fig1 seems to suggest that Cxcl5 is expressed in AH livers. What cell type in the liver expresses Cxcl5?

The neutrophil recruitment assay shown in supplementary figure 8 suggest that conditioned medium from LSECs stimulate neutrophil recruitment. Unclear if this conditioned medium was from LSECs treated with TNFa. Is there a difference between LSEC conditioned medium from LSECs

treated with TNFa and TNFa+Celastrol?

Figure 3, panel A. What does HHSEC refer to? Wasn't the 4C experiment performed in LSEC with and without TNFa?

The authors exploit previous published H3K27Ac, BRD4 and NF-kB ChIP-seq data in HUVECs and translates this data to LSECs. Although the data correlate well with LSECs at the Cxcl locus the authors should at least validate the super-enhancer in LSECs (-/+ TNFa) using H3K27Ac ChIP-seq. Since the overall study argues that this is a super-enhancer in LSECs it will be important to confirm this.

According to supplementary figure 9B TNFa treatment of HUVEC cells does not lead to a pronounced change of H3K27Ac at the entire super enhancer although NF-kB is recruited. Some change is seen near the Cxcl8 gene which could indicate that an enhancer near Cxcl8 is the primary regulator and not necessarily by the so-called super-enhancers. Is this also observed in LCES? Refers back to the question above. Is there a super-enhancer near Cxcl8 in LSECs? Super-enhancer or not, this enhancer region still seems important for expression of associated genes. But the super enhancer definition may be questionable and if the ROSE algorithm does not find a super enhancer the super enhancer term should be rephrased throughout the manuscript.

"Cells receiving sgRNA treatment only without dCas9-KRAB showed no change in expression of CXCL genes (data not shown)." This is an important control and the data should be shown in a supplementary figure.

The 3C experiment shown in figure 3F suggests that the Cxcl5,3 and 2 genes are not connected to the Cxcl8 proximal enhancer. Does the gCas9-KRAB targeting this enhancer effect Cxcl5,3 and 2 expression?

Along the same line as above. Does treatment with BET inhibitors (pan and specific BET inhibitors) affect Cxcl5,3 and 2 expression?

In the paragraph from line 311-337 on page 9 the authors argue that the Cxcl enhancer contacts Cxcl8,6 and 1 genes in macrophages. Macrophages are recruited to the AH diseased liver, yet the authors suggest that the Cxcl genes are primary expressed by LSECs. Why are the Cxcl genes not expressed by liver resident macrophages during the inflammatory response in AH?

To test if the two in vitro tested BET inhibitor could reduce the hepatic inflammatory response in a mouse model for AH the authors use a previously described experimental setup (Figure 6 and 7). They show increased expression of Cxcl1 and Cxcl2 in whole liver. In the mouse model is Cxcl1 and Cxcl2 expression predominantly from LSECs or macrophages or other liver resident cell types? In other words what is the primary target of the BET inhibitors in the mouse model?

Minor comments:

Supplementary figure 3A. Plot Histone mods as z-score

On page 9 line 303 the description of the data refers to figure supp. Figure 11. This should be Figure 5B.

On page 9 line 304 the description of the data refers to figure 5B. This should be supp. Figure 11.

Dear Reviewers,

Thanks very much for your comments. We value your feedback and have attempted to address each question brought to our attention. Please refer to our point-by-point response letter below with references. We hope that our revised contents and responses will satisfy Reviewer concerns and support our conclusions.

REVIEWER COMMENTS

Reviewer #1 (Remarks to the Author):

In this paper Liu et al study the mechanistic role of super-enhancers in chemokine expression in the context of alcoholic hepatitis. Combining studies in human liver samples, *in vitro* studies and *in vivo* studies in mice exposed to bromodomain inhibitors, they show that epigenomic changes in livers from patients with alcoholic hepatitis is associated with up regulation of CXCL chemokines. They identify super-enhancer to CXCL chemokines upstream of the CXCL locus, in particular in liver endothelial cells, that govern CXCL expression, as shown when silencing this super-enhancer by CRISP dCas9-KRAB or by pharmacological bromodomain inhibitors. Finally, using murine models of alcoholic liver disease with more or less severity, they demonstrate that BET inhibitors reduce liver expression of CXCL and infiltration of neutrophils. The study, performed by a well established group in the field, is interesting and novel, and contains an impressive amount of data. However, I have several major concerns, in particular regarding *in vivo* studies, that need to be addressed and found some data overinterpreted.

We thank the reviewer for these comments. We have amended the manuscript to address these concerns; they are addressed in greater detail in point by point responses below.

1. The reason as to why the study on isolated cells was exclusively focused on endothelial cells is not clear and relies on data from normal liver excluding immune cells. The fact that LSEC is the predominant cell type expressing CXCL chemokines in the normal liver is not a sufficient argument to exclude immune cells which are a major source of inflammatory mediators, most likely including CXCL in AH, as underlined in Sup Fig 13 (now **Supp Figure 21**). This is of importance since in *in vivo* studies, pharmacological inhibitors will not only target endothelial cells.

Thank you for this important comment related to sources of chemokines in liver diseases¹⁻³. Bulk RNA-seq of isolated LSECs, as well as recent single cell RNA sequencing (scRNA-seq) studies of normal livers in human and mouse, have found LSECs to be important sources of CXCL chemokines (**Supp Table 1**). In AH, TNF α has a central role in orchestrating immune responses, and LSECs upregulate CXCL chemokine expression to TNF α stimulation *in vitro* as demonstrated in this study and *in vivo* in recent publications^{4,5}. LSEC was shown to upregulate *Cxcl1* within 2 hours of TNF α treatment in mice liver⁴. Another study demonstrated that with Kupffer cell depletion, LSECs and HSCs were the primary sources of chemokine production, including *Cxcl1*, in response to cytokines, TNF α and IL-1⁵. These studies highlighted the ability of LSECs to activate in early inflammation and propagate the inflammatory cascade in a process of “angiocrine signaling”, which describes endothelial paracrine signaling that regulates many pathobiologic processes⁶. We recognize the importance of immune cells in chemokine production. Whereas LSEC is important in early inflammation, recruited immune cells amplify inflammatory cascade at later stages. Our study and references in **Supp Table 1** highlight the plurality of cellular sources that contribute to chemokine production, and the same TNF α -responsive super-enhancer may regulate chemokine production in multiple cell types and can be suppressed with pharmacologic BET protein inhibition. In this study, we aim to use LSEC as a platform to highlight a valuable molecular and cellular target for therapeutic intervention in AH. We have incorporated additional discussion of cellular sources of chemokines in lines 188-190, 432-447, 456-460, and 478-495 of the main text.

2. In *in vivo* studies, reduction of CXCL expression is shown in two models of alcohol feeding. Nevertheless, a major issue in the context of the disease is to reduce inflammation but also in liver injury. However, the authors limit their studies to chemokine levels quantification and MPO. Strikingly, they do not investigate steatosis (which surprisingly when looking at IHC is not obvious), and provide results on ALT that are either not convincing or come from a very low number of animals. Caspase and oxidative stress should also be assessed.

We thank the reviewer for this valuable comment which has been addressed by increasing the sample size and scope of our analyses. We now include new data examining caspase activation and oxidative stress. We demonstrate that cleaved caspase 3 is increased with alcohol/LPS treatment, and that this increase is attenuated by iBET151 (Supp Figure 26a). We also demonstrate similar levels of lipid peroxidation and DNA oxidation as assessed by measurements of malondialdehyde (MDA) and 8-hydroxy-2-deoxy Guanosine (8-OHdG) from liver of mice in the experimental groups (Supp Figure 26b, c). We now demonstrate increased steatosis in alcohol fed/LPS mice in the 2-week feeding model with attenuation in mice given iBET151 (Supp Figure 25). There is no difference in liver steatosis in mice in the 3-day alcohol binge model (Supp Figure 27). The anti-steatotic effect from BET inhibition may reflect direct effects on lipogenesis and was recently observed in a NASH model of liver injury^{7,8}. No significant change was observed in ALT levels among treatment groups (Figures 6d and 7d). Our new data have been incorporated into lines 385-392, 526-531, and 539-543 of the main text.

3. In *in vivo* studies, the number of samples analysed vary with the parameter studied. For example in Fig 6, MPO results are n=4 for ETOH LPS, but n=16 for Ly6G, or n=14 for CXCL. In sup Fig 14 (now Supp Figure 23) outliers were excluded for CXCL, and ALT data are analysed in n=4 for ETOH and more for CXCL etc...6-8 pair fed are indicated in the legend and 12-14 alcohol?. Why do the authors do not show the results of all treated animals and the number shown vary with parameters? Same comments for Fig 7, for example for ALT

We thank the reviewer for this critique. We now analyze and present all samples for MPO, *Ly6g*, *Cxcl1*, *Cxcl2* and ALT in Figures 6 and 7.

4. The main effect of sg RNA in Fig 4B and C is to reduce basal expression of CXCL. Viability tests showing the lack of toxicity should be provided. Same comments for bromodomain inhibitors in Fig 5.

We now utilize a live cell imaging system, IncuCyte, (Essen BioScience)⁹ to study cytotoxicity of CRISPR cells and bromodomain inhibitors. Cells expressing sgRNA/dCas9 constructs did not show increased cytotoxicity compared to control (Supp Figure 15). Bromodomain inhibitor cytotoxicity studies were focused on iBET151 as it is better authenticated than UMN627. This showed no increased cytotoxicity at concentration used in the study from 1uM to 50uM (Supp Figure 19). We have incorporated this into lines 286-288 and 317-319 of the main text.

Additional comments.

Which CXCL is assessed by ELISA in Fig 2B?

CXCL1 was assessed by ELISA.

In exp with LSEC medium provided in sup Fig 8, it looks like if LSEC are dying ?

We addressed the concern of LSEC viability by using a Transwell assay for neutrophil chemotaxis studies in which LSEC viability was much better. We now demonstrate that LSEC supernatant promotes neutrophil chemotaxis, which was further enhanced by pretreatment with TNF α and inhibited by pharmacologic NF- κ B inhibition with Celastrol (Supp Figure 9). We have incorporated this result into lines 207-209 and 213-215 of the main text.

In Fig 5A for each chemokine, data shown are - and + TNF?

Thank you. We revised the labeling for clarity.

Reviewer #2 (Remarks to the Author):

Comments for the authors

The studies in this paper use a variety of sophisticated methods to examine the changes in CXCL expression and underlying regulatory mechanisms in liver samples from patients with alcoholic hepatitis and the potential therapeutic effectiveness of bromodomain inhibitors for alcoholic hepatitis in two mouse models. The major advances of the study are: 1) the identification of sinusoidal endothelial cells as a major contributor of CXCL expression; 2) sophisticated confirmation of the long range interactions of an enhancer region regulating multiple CXCL genes upstream of the genes; and 3) the demonstration of the effectiveness of pan-bromodomain inhibition or specific BD1 inhibition in attenuating alcoholic hepatitis in mouse models. However, there are significant concerns. Evidence for the specific role of alcohol in the regulatory mechanism proposed is not presented, so it is not clear whether alcoholic hepatitis or general liver inflammation and TNF α /NF- κ B signaling is being studied. The epigenetic studies are weak and not complete and evidence for epigenetic control and a “super” enhancer is weak. The roles of a super-enhancer regulating CXCL expression in mouse and humans and CXCL involvement in alcoholic hepatitis are not novel.

We thank the reviewer for his/her careful review and thoughtful comments. We have performed additional experiments to directly address the role of alcohol feeding in our *in vivo* model, which is described in the point-by-point responses below (**Supp Figure 24**). We have also provided additional analysis of human AH RNA-seq studies to address the potential roles of epigenetic modifiers in gene regulation in AH. We hope these new additions will strengthen and support our conclusions.

1. The studies with the KRAB construction to demonstrate epigenetic control is not compelling. The control dCas9 without KRAB has a similar, but smaller effect suggesting that another mechanism is involved, perhaps steric hindrance as mentioned, which might be enhanced by the additions of the KRAB to the dCas9. Evidence for epigenetic control is therefore weak.

We thank the reviewer for this comment. To further investigate the potential role of epigenetic regulation, we performed new H3K27ac ChIP-seq in LSECs. We found an enrichment of H3K27ac mark at the CXCL SE and promoter sites after TNF α stimulation (**Supp Figure 11**), which mirrors the increased H3K27ac occupancy seen in AH livers. With addition of NF- κ B inhibitor, H3K27ac augmentation to TNF α stimulation is attenuated, highlighting possible epigenetic regulatory control by histone acetylation (**Supp Figure 12b**). We agree with the reviewer that steric hinderance likely plays a role in dCas9-KRAB gene repression at target sites given the smaller but significant effect by dCas9 control construct at the same targets. However, steric interference of transcription factor at target site will likely also decrease histone acetylation mediated by binding of various transcription coactivators. In total, we believe that histone mark editing plays a role in transcriptional regulation of the CXCL locus but is only part of the mechanism. We have incorporated additional discussion regarding epigenetic control into the main text, lines 258-264, 305-309, and 485-489.

2. If the emphasis of the study is on epigenetic studies, then, the important epigenetic modifiers that are altered in alcoholic hepatitis should be identified.

This is a valuable suggestion as our IPA analysis has identified multiple epigenetic modifier pathways to be differentially activated in AH, including SP1 and SMARCA4 which were amongst top 10 transcription regulators with the highest activation z-scores (**Figure 1e**). Others such as SMARCD3, BCL6 and TBXT were also ranked highly. Among these epigenetic modifiers, SMARCA4 and SMARCD3 are both members of the SWI/SNF (SWItch/Sucrose Non-Fermentable) family of ATP-dependent chromatin remodeling complexes, which regulate access to the chromatin, allowing genes to be activated or repressed¹⁰. SP1 is a transcription factor that also interacts with histone acetyltransferases p300 and cooperatively regulates gene expression¹¹. Moreover, NF- κ B signaling pathway is not only a top-activated pathway in AH, it is also well known as an important recruiter of p300, which mediates histone acetylation¹² (**Supp Figure 12b**) and

is important in liver fibrogenesis, as we demonstrated in a recent publication¹³. This discussion is incorporated into lines 166-170, 489-495 of the main text.

3. Relative to the KRAB studies, if this is a super-enhancer with multiple NF- κ B sites, why would targeting a single NF- κ B site have such a dramatic effect, i.e. reducing the effectiveness of the entire complex of enhancers. Was there a change in the epigenomic landscape of the entire super-enhancer by targeting one NF- κ B site?

We thank the reviewer for this astute observation. We indeed observed a very robust effect on CXCL gene expression from a single sgRNA targeting one NF- κ B binding site in the SE. Multiple sgRNAs (15 in total) were designed targeting each of the 4 strongest predicted NF- κ B binding peaks in the CXCL SE, and the sgRNA (E1) with the strongest activity was selected for further studies (**Supp Figure 14**). Of the several NF- κ B binding peaks identified in NF- κ B ChIP-seqs, the highest peak identified in the CXCL SE region was also the target of sgRNA E1, which had the best activity among sgRNAs from our experimental testing. Therefore, the NF- κ B binding site targeted by E1 was likely the most consequential in driving CXCL SE activity. We assessed whether chromatin interactions changed with dCas9-KRAB targeting or TNF α stimulation and contributed to transcriptional changes. While TNF α enhances interaction of super-enhancer with multiple CXCL promoters (**Figure 3f**), chromatin interactions were not significantly decreased after KRAB targeting at the SE site (**Supp Figure 13**). The reviewer raised another possibility that the dCas9-KRAB repression on the target site may change the epigenomic landscape of the entire super-enhancer. However, if the effect of dCas9-KRAB mediated histone change were broad, we would expect the effects of different sgRNAs targeting the CXCL SE to be similar, which was not seen. Prior publication has implicated that dCas9-KRAB induced H3K9me3 deposition generally affects chromatin up to 4.5 kb away from the target site¹⁴, which would be a very limited distance compared to the size of the SE (75 kb) (**Supp Figure 7**). We have expanded our discussion of dCas9-KRAB targeting in the main text lines 245-249, 270-272, 282-286, and 302-303.

4. As noted by the authors, a super-enhancer regulating CXCL expression in mice and humans and a role of CXCL in alcoholic hepatitis have been previously described, which reduces novelty of the findings in this paper.

To our knowledge, this is the first study linking super-enhancer regulation to human liver disease pathogenesis aside from liver malignancies. This study also utilizes human AH tissue to characterize histone epigenomic changes in AH, which is understudied. The role of super-enhancer gene regulation in AH and underlying mechanisms have not been studied prior to this manuscript. Importantly, our study is the first to show CRISPR-dCas9-KRAB targeting of the CXCL super-enhancer or pharmacologic BET domain inhibitors can successfully inhibit multiple CXCLs expression, providing novel, druggable targets for future investigations. This study significantly extends our understanding of AH and therapeutic implications for treatment.

5. What is being studied in the mouse models is confounded by the stimulation with LPS. It is not clear whether the changes in regulation of *Cxcl* are due to alcohol toxicity or to general inflammation with LPS. Comparison of alcohol + LPS with LPS might allow changes specific to alcoholic hepatitis to be identified.

We thank the reviewer for this insightful comment. Alcohol feeding models, including the NIAAA chronic-binge model, accurately mimic alcoholic steatosis but have very modest effects on inflammation (**Supp Figure 23**)¹⁵. To enhance the inflammatory reaction to liver injury in these mice, we combined alcohol feeding with LPS, which is an important molecule in the pathogenesis of AH¹⁶. Compared to LPS injection alone, combination of alcohol feeding and LPS treatment increased *Cxcl1* and *Cxcl2* expression and significantly increased steatosis (**Supp Figure 24**), supporting its use as a validated model of AH^{17,18}. We have incorporated this result in the main text lines 376-380 and 526-531.

6. Data presentation could be improved.

- Box and whisker plots are useful for larger datasets, but n values in most of these experiments are small and all points should be plotted as in Fig. 6 and 7.

We thank the reviewer for this suggestion. The graph format has been changed to scatter plots.

- Also, using log plots are useful if there are order of magnitude differences, but linear plots are more intuitive and better for less than 100-fold differences, which is the case for most of these plots that use the log scale.

We thank the reviewer for this suggestion. The graph format has been changed to a linear scale for all ChIP-qPCR data as well as all *in vivo* data (Figure 2a-b, Figure 3c, Figure 4d, Figure 5c-f, Figure 6, Figure 7, Supp Figure 6, Supp Figure 8c, Supp Figure 9, Supp Figure 12, Supp Figure 23-28). In assays involving BET inhibitors, CRISPR cells and macrophage expression, log scale graphs were retained to better highlight fold changes >100 and changes at basal condition (Figure 2c, Figure 4b-c, Figure 5a-b, Supp Figure 14, Supp Figure 16, Supp Figure 18, Supp Figure 22).

- Fig. 4D, plots use derived data of fold difference and normalization to control for a ChIP-seq. Better to plot more direct data as % input as in Fig. 3C and 5C-F.

We thank the reviewer for this suggestion. The graph format has been changed.

Minor

L61 - delete "on"

We thank the reviewer for his/her careful review. We have changed "acute on chronic liver failure" to "acute-on-chronic liver failure", which is a term used to describe acute insult or injury to the liver in the background of chronic liver disease¹⁹.

L182 - prorogating = propagating

L217 - chemokine - chemokines

L322, 325, 327 - macrophage = macrophages

L331 - expressions = expression and folds = fold

L448 - additional = addition

L487 - simulate = simulates

We thank the reviewer for her/his careful review. We have now corrected these errors.

L231-234 - BRD4 is analyzed in Supp. Fig. 9B and should be mentioned here.

We thank the reviewer for her/his careful review. We have discussed BRD4 ChIP-seq result here as suggested, lines 242-244.

Fig. 5C-F - indicating the antibody of control IgG used with + and - below the graph can be confusing suggesting for example that BRD4 in C is expressed in the cells. Better to indicate BRD4, etc. and IgG above the bar plots.

We thank the reviewer for this suggestion. The graph format has been changed (Figure 5c-f).

L343 - In Supp. Fig. 14 (now Supp Figure 23), Cxcl2 expression did not increase contrary to the statement in the text. Also Ly6g which is measured in this fig should be mentioned

We thank the reviewer for this suggestion. The text has been edited to more accurately reflect the results, lines 365-367.

L355 - BODIFY staining for steatosis in Supp. Fig. 15 (now Supp Figure 25) is increased as described, but there is no mention of Oil Red O staining in Supp. Fig. 14 (now Supp Figure 23).

We thank the reviewer for this suggestion. The text has been edited in lines 369-370.

L361 - ALT levels and Steatosis did not show statistically significant differences with iBET151 as stated, but this is a little misleading, suggesting no difference. The data show a decrease trend that was not significant (Fig. 6D and Supp. Fig. 15-now **Supp Figure 25**)

We thank the reviewer for this comment. We analyzed more samples and now demonstrate a significant reduction in steatosis in EtOH fed/LPS mice receiving iBET151. We have updated the main text as well as **Supp Figure 25** to reflect this change in line 385-386 and 539-543. Changes in ALT levels were not significant.

Reviewer #3 (Remarks to the Author):

The manuscript by Liu and colleagues describes a study that identifies mechanisms regulating chemokine production from the Cxcl locus in human liver from patients suffering from alcoholic hepatitis. The authors first use a genomics approach (a combination of RNA-seq and ChIP-seq against various histone modifications) to find differentially regulated genes and enhancers in human liver biopsies from healthy and diseased donors. Among TNF α and NF- κ B controlled inflammatory pathways, they find differential regulation of Cxcl1,6 and 8 expression. Using a combination of publicly available data and gene expression analysis in different hepatic cell types they argue that Cxcl1,6 and 8 are predominantly expressed in human liver sinusoid endothelial cells (LSECs). Based on genome wide H3K27Ac in human umbilical vein endothelial cells (HUVECs) and chromatin conformation assays (3C and 4C-seq) in LSECs they suggest that Cxcl1,6 and 8 expression is controlled by a super-enhancer in close vicinity to Cxcl8. The enhancer is occupied with NF- κ B and BRD4 in response to TNF α in LSECs in line with differential Cxcl1,6 and 8 expression upon an inflammatory response. Using Cas9-KRAB and gRNA against the putative super-enhancer they convincingly demonstrate that the enhancer is needed for maximal expression of Cxcl1,6 and 8. Lastly, they demonstrate that BED inhibitors can attenuate Cxcl1,6 and 8 expression and lower the inflammatory response in a mouse model of alcoholic hepatitis.

Identification of mechanisms controlling neutrophil recruitment to the inflamed liver is of high interest in order to develop targeted anti-inflammatory strategies in tissue such as the liver. This study suggests a novel mechanism for cytokine induced expression Cxcl1,6 and 8 and based on this suggests an anti-inflammatory strategy targeting regulatory enhancer activity by BED inhibitors. Their study is overall well designed, and the individual experiments are sound. However, there are a couple of concerns relating specifically to the LSECs as the major contributor to Cxcl1,6 and 8 expression in the inflamed liver and the suggestion that expression is controlled by a so-called super-enhancer.

We thank the reviewer for his/her careful review and thoughtful comments. These points are addressed in detail below.

Major concerns:

1. The authors perform a fairly robust analysis of RNA-seq and histone mod ChIP-seq. However, details on sequencing depth and uniquely aligned reads/fragments for the individual libraries is not provided. This should at least be listed in a table so the reader can evaluate the NGS data quality.

We thank the reviewer for the positive comments and suggestions. We have attached two tables to show details of the sequencing depth and uniquely aligned reads/fragments for the individual libraries for NGS data quality evaluation (**Supp Table 2 and 3**).

2. Cell type specific expression of Cxcl's was first assessed by mining the FANTOM5 database and confirmed using RNA-seq in different primary human liver cell types. Unclear from the material and methods how these primary cell types were isolated from human liver. Should be clarified in the material and methods. Also, to further validate the cell type specific expression of Cxcl's the authors should show expression of relevant Cxcl's from previously published human liver single cell data (PMID: 31292543, PMID: 30348985 and PMID: 31597160). This will specifically address which cell types of the liver that express the different Cxcl genes.

We thank the reviewer for this suggestion. Primary liver cells used in our bulk RNA-seq study as well as the one in the FANTOM5 database were purchased from ScienCell. The information about cell isolation methods by ScienceCell has been added to Supplemental Methods (in Supp Materials), line 378-387.

We appreciate the reviewer's suggestion that we examine published scRNA-seq data to further delineate the cell sources of CXCL chemokines. Numerous published scRNA-seq studies of human and mouse liver cells have been reviewed in preparation of this revision, and the cellular sources of CXCL chemokines are summarized in **Supp Table 1**. LSECs are frequently among the top 3 cell types of chemokine production in the liver. Furthermore, LSECs upregulate CXCL chemokine in response to TNF α signaling not only *in vitro*, but also *in vivo*^{4,20}. As relatively abundant resident cells, LSECs are the second largest cell population next to hepatocytes and higher than Kupffer cells²¹⁻²³. LSECs are important in recruiting immune cells early in the inflammatory response when the condition can be reversed. Subsequent infiltrating immune cell CXCL chemokine production may be important for amplification of the inflammatory cascade at the later times that the disease has already progressed significantly. We have incorporated additional discussion of cellular sources of chemokines in the main text, lines 188-190, 432-447, 456-460, and 478-495.

3. The illustration of Cxcl locus shown in supplementary figure 7 could be improved. Would be informative to show the locus with relative distance between the genes and enhancers. Here a browser shot from for example the UCSC genome browser would be helpful. Here H3K27Ac could also be shown to highlight the position of the super-enhancer. Also, the indicated predicted NF- κ B binding sites should be shown.

We thank the reviewer for this valuable suggestion. We have amended **Supp. Figure 7**, and the H3K27Ac ChIP-seq in LSEC was used as a template to show relative distances between genes of interest and CXCL SE. NF- κ B binding sites investigated in this study were highlighted as well.

4. Experiments using TNF α and a NF- κ B inhibitor treatment of primary LSECs shown in figure 2B and C suggest that NF- κ B is involved in Cxcl1,6 and 8 expression in LSECs. This may be directed by NF- κ B binding to the promoters and the super-enhancer indicated in Suppl. Figure 7. Would be informative to show NF- κ B recruitment to the predicted NF κ B binding sites at promoters and super-enhancer using ChIP. Although the NF- κ B ChIP is shown for one site (super-enhancer) in figure 3C, it would be informative to probe NF- κ B at all suggested binding sites. Also is H3K27Ac at the promoters, enhancers and super-enhancer changed in response to TNF α treatment and inhibited by Celestrol? This would indicate if promoter and enhancer activity is regulated by TNF α and NF- κ B.

We agree with the reviewer that TNF α stimulation increased NF- κ B binding to CXCL genes and CXCL SE. We performed new ChIP-qPCR experiments to assess for NF- κ B binding at predicated binding sites in the CXCL promoter regions and in CXCL SE as illustrated in **Supp Fig 7**. NF- κ B binding is indeed enriched at these sites and is increased with TNF α stimulation (**Supp Figure 12a**). We have also performed new ChIP-seq studies to address this query. NF- κ B ChIP-seq of LSEC demonstrated increased NF- κ B binding after TNF α stimulation at multiple predicted binding sites similar to previous observation in HUVEC cells²⁴. The NF- κ B binding site targeted by the most effective sgRNA (E1) was the highest NF- κ B binding peak in the CXCL SE in both LSEC and HUVEC ChIP-seq studies (**Supp Figure 10b, 11a**). We also performed new H3K27ac ChIP-seq in LSECs, which demonstrated increased H3K27ac occupancy with TNF α treatment at the CXCL promoters and SE (**Supp Figure 11a**). We have similarly shown this with new H3K27 ChIP-qPCR (**Supp Figure 12b**). In addition, we showed that TNF α mediated H3K27ac enrichment was attenuated by Celestrol, which suppressed NF- κ B/BRD4 binding at these sites as well as CXCL gene expression (**Supp Figure 12b**). NF- κ B/BRD4 binding appears necessary for TNF α mediated H3K27ac deposition at target sites. We have incorporated the result of the ChIP-seq as well as ChIP-qPCR results into the main text lines 237-240, 258-260, and 305-309.

5. The Cxcl2, 3 and 5 are in vicinity of the Cxcl1,6 and 8 genes. Are Cxcl2, 3 and 5 expression changed by TNF α treatment in LSECs? Also are Cxcl2, 3 and 5 differentially regulated by AH in liver? Fig1 seems to suggest that Cxcl5 is expressed in AH livers. What cell type in the liver expresses Cxcl5?

We thank the reviewer for this question. *In vitro*, expression of *CXCL2*, *3*, and *5* is enhanced with TNF α treatment in LSEC (Supp Figure 16a, 16b). In AH livers, *CXCL2* expression is downregulated, consistent with prior studies²⁵. In humans, *CXCL2* is expressed at relatively high levels in hepatocytes as evidenced by FANTOM data (Supp Figure 6) and scRNA-seq²⁶. Hepatocyte injury in AH leads to dysregulation of many hepatocyte genes that may contribute to these disparities. *CXCL3* and *CXCL5* are minimally expressed in our patient RNA-seq samples compared to other chemokines, with *CXCL5* upregulated in AH and no change to *CXCL3*. scRNA-seq suggests *CXCL3* and *CXCL5* are primarily produced by cholangiocytes and Kupffer cells at low levels in normal liver tissue²⁶⁻²⁸ (Supp Table 1).

6. The neutrophil recruitment assay shown in supplementary figure 8 suggest that conditioned medium from LSECs stimulate neutrophil recruitment. Unclear if this conditioned medium was from LSECs treated with TNF α . Is there a difference between LSEC conditioned medium from LSECs treated with TNF α and TNF α +Celastrol?

We thank the reviewer for this question. We have performed additional experiments studying the effect of inhibitor Celastrol on neutrophil chemotaxis. Using a Transwell neutrophil migration assay, we demonstrated that neutrophil chemotaxis was enhanced in the presence of LSEC cultured medium, which was further enhanced with TNF α pretreatment and diminished after addition of Celastrol (Supp Figure 9), lines 207-209 and 213-215.

7. Figure 3, panel A. What does HHSEC refer to? Wasn't the 4C experiment performed in LSEC with and without TNF α ?

We thank the reviewer for his/her careful review. We have changed the inconsistent nomenclature. We have now uniformly used LSEC to refer to liver sinusoidal endothelial cells throughout the paper.

8. The authors exploit previous published H3K27Ac, BRD4 and NF-kB ChIP-seq data in HUVECs and translates this data to LSECs. Although the data correlate well with LSECs at the Cxcl locus the authors should at least validate the super-enhancer in LSECs (-/+ TNF α) using H3K27Ac ChIP-seq. Since the overall study argues that this is a super-enhancer in LSECs it will be important to confirm this.

We thank the reviewer for this suggestion, and we agree that confirming the presence of CXCL SE in LSEC would be critical to the study. We have performed new H3K27ac ChIP-seq on LSEC with or without TNF α stimulation in duplicates (Supp Figure 11a) (GSE166564). Indeed, we found that H3K27ac was strongly enriched in the predicated SE area. ROSE algorithm identified this region to be a super-enhancer in LSEC, and its ranking rose after TNF α stimulation (from 53 to 11 percentile in the ~500 SE identified) (Figure. 3c). We have therefore confirmed the presence of CXCL SE in LSEC and have updated Figure 3 to illustrate this finding. We have incorporated discussion of the CXCL SE in the main text, lines 258-264.

9. According to supplementary figure 9B TNF α treatment of HUVEC cells does not lead to a pronounced change of H3K27Ac at the entire super-enhancer although NF-kB is recruited. Some change is seen near the Cxcl8 gene which could indicate that an enhancer near Cxcl8 is the primary regulator and not necessarily by the so-called super-enhancers. Is this also observed in LCES? Refers back to the question above. Is there a super-enhancer near Cxcl8 in LSECs? Super-enhancer or not, this enhancer region still seems important for expression of associated genes. But the super-enhancer definition may be questionable and if the ROSE algorithm does not find a super-enhancer the super-enhancer term should be rephrased throughout the manuscript.

We thank the reviewer for their comments. Unlike in HUVEC cells where TNF α treatment did not noticeably increase H3K27ac occupancy²⁴, TNF α treatment did increase H3K27ac occupancy in LSEC.

We agree that it is important to confirm the presence of the CXCL SE in LSEC cells, which we have now done (**Figure 3d-e, Supp Figure 11a**), lines 258-264. The CXCL SE was identified by ROSE algorithm in both LSEC and HUVEC, and its ability to regulate multiple CXCL chemokines further supports this finding.

10. “Cells receiving sgRNA treatment only without dCas9-KRAB showed no change in expression of CXCL genes (data not shown).” This is an important control and the data should be shown in a supplementary figure.

We agree with the reviewer and have included additional data on sgRNA alone treatment in the **Supp Figure 16c, 16d**. There is no effect on CXCL expression with sgRNA treatment alone.

11. The 3C experiment shown in figure 3F suggests that the *Cxcl5,3* and 2 genes are not connected to the *Cxcl8* proximal enhancer. Does the sgCas9-KRAB targeting this enhancer effect *Cxcl5,3* and 2 expression? Along the same line as above. Does treatment with BET inhibitors (pan and specific BET inhibitors) affect *Cxcl5,3* and 2 expression?

We thank the reviewer for this question. We have updated 3C experiments to include conditions with and without $TNF\alpha$ stimulation, and with increased sample size and find increased interaction of CXCL SE with *CXCL 1, 2, 3, 6, and 8* (**Figure 3f**). There is also increased SE/promoter interaction with $TNF\alpha$ stimulation (**Figure 3f**). We agree with the reviewer that it would be important to assess the response of *CXCL2, 3, and 5* (**Supp Figure 16a, 16b**) to CXCL SE suppression. In LSECs, *CXCL2, 3, and 5* production increases with $TNF\alpha$ stimulation. The response to CXCL SE suppression by dCas9-KRAB is less robust compared to *CXCL 1, 6, and 8*. There was a trend toward decreased *CXCL3* and *CXCL5* expression after KRAB mediated SE suppression, but these changes were not statistically significant. BET inhibitor treatment with iBET151 or UMN627 resulted in decreased expression of *CXCL2, 3, and 5* (**Supp Figure 18a**). It is possible that the regulatory effect of the CXCL SE may be weaker with *CXCL2, 3, and 5* compared to other chemokines, possibly due to the relative proximity of *CXCL 1, 6, and 8* to the CXCL SE compared to *CXCL2, 3, and 5* (**Supp Figure 7**). The higher potency of CXCL suppression with BET inhibitors, which are effective for the whole length of the CXCL SE, compared to dCas9-KRAB treatment, which only targets one NF- κ B binding site, may be due to the breadth of treatment. We have incorporated results regarding *CXCL2, 3, and 5* regulation into the main text, 288-290.

12/13. In the paragraph from line 311-337 on page 9 the authors argue that the *Cxcl* enhancer contacts *Cxcl8,6* and 1 genes in macrophages. Macrophages are recruited to the AH diseased liver, yet the authors suggest that the *Cxcl* genes are primarily expressed by LSECs. Why are the *Cxcl* genes not expressed by liver resident macrophages during the inflammatory response in AH? To test if the two in vitro tested BET inhibitor could reduce the hepatic inflammatory response in a mouse model for AH the authors use a previously described experimental setup (Figure 6 and 7). They show increased expression of *Cxcl1* and *Cxcl2* in whole liver. In the mouse model is *Cxcl1* and *Cxcl2* expression predominantly from LSECs or macrophages or other liver resident cell types? In other words what is the primary target of the BET inhibitors in the mouse model?

We thank the reviewer for these insightful questions about cellular sources of CXCL chemokines. In mice, scRNA-seq studies have found that both LSEC and macrophages are major sources of *Cxcl1* (expression levels of *Cxcl2* were quite low; **Supp Table 1**). For example, LSECs were the top source of *Cxcl1* production in a scRNA-seq study on liver nonparenchymal cells in mice. In terms of macrophage, while peripheral blood monocytes have relatively low levels of CXCL expression and are generally unresponsive to canonical stimulation to enhance CXCL production²⁹, after differentiation into macrophages, we now show that these cells do indeed upregulate CXCL production in response to LPS and $TNF\alpha$ stimulation (**Supp Figure 22**). While both cells produce *CXCL 1*, in a disease context, LSEC as resident cells are critical for the early injury response that recruits immune cells. We certainly recognize the role of immune cells in chemokine production in inflammation, but this occurs at later stages when the disease process is already

advancing. This “angiocrine signaling” process is increasingly recognized in diverse disease processes whereby endothelial cells regulate broader organ pathophysiologic processes⁶. Our new studies and those we cited in **Supp Table 1**, indicate that the same SE may be responsible for chemokine regulation in multiple cell types which can be blocked by BET inhibitors. Thus, the primary *in vivo* target of BET inhibitors is likely to be LSEC as well as macrophages; however, LSEC are the better target in the critical and early stages of the disease process. We have incorporated additional discussion of macrophage CXCL chemokine expression in the main text, lines 347-351.

Minor comments:

14. Supplementary figure 3A. Plot Histone mods as z-score

We thank the reviewer for this suggestion. The graph format has been changed.

15. On page 9 line 303 the description of the data refers to figure supp. Figure 11 (now removed). This should be Figure 5B.

On page 9 line 304 the description of the data refers to figure 5B. This should be supp. Figure 11 (now removed).

We thank the reviewer for detecting this. **Figure 5b** refers to *in vitro* UMN627 inhibition of CXCL expression. The previous **Supp Figure 11** refers to properties of UMN627, and since initial submission of the manuscript, relevant studies on UMN627 have been published. Previous **Supp Figure 11** is now removed for brevity and replaced by a citation³⁰. We have edited the main text for clarity, lines 321-324.

REFERENCES:

- 1 Sahin, H., Berres, M. L. & Wasmuth, H. E. Therapeutic potential of chemokine receptor antagonists for liver disease. *Expert Rev Clin Pharmacol* **4**, 503-513, doi:10.1586/ecp.11.24 (2011).
- 2 Marra, F. & Tacke, F. Roles for Chemokines in Liver Disease. *Gastroenterology* **147**, 577-594.e571, doi:<https://doi.org/10.1053/j.gastro.2014.06.043> (2014).
- 3 Girbl, T. *et al.* Distinct Compartmentalization of the Chemokines CXCL1 and CXCL2 and the Atypical Receptor ACKR1 Determine Discrete Stages of Neutrophil Diapedesis. *Immunity* **49**, 1062-1076 e1066, doi:10.1016/j.immuni.2018.09.018 (2018).
- 4 Krausgruber, T. *et al.* Structural cells are key regulators of organ-specific immune responses. *Nature* **583**, 296-302, doi:10.1038/s41586-020-2424-4 (2020).
- 5 Bonnardel, J. *et al.* Stellate Cells, Hepatocytes, and Endothelial Cells Imprint the Kupffer Cell Identity on Monocytes Colonizing the Liver Macrophage Niche. *Immunity* **51**, 638-654 e639, doi:10.1016/j.immuni.2019.08.017 (2019).
- 6 Rafii, S., Butler, J. M. & Ding, B.-S. Angiocrine functions of organ-specific endothelial cells. *Nature* **529**, 316-325, doi:10.1038/nature17040 (2016).
- 7 Middleton, S. A. *et al.* BET Inhibition Improves NASH and Liver Fibrosis. *Sci Rep* **8**, 17257, doi:10.1038/s41598-018-35653-4 (2018).
- 8 Gilan, O. *et al.* Selective targeting of BD1 and BD2 of the BET proteins in cancer and immuno-inflammation. *Science*, eaaz8455, doi:10.1126/science.aaz8455 (2020).
- 9 O'Clair, L. *et al.* Quantification of cell migration and invasion using the IncuCyte™ Chemotaxis assay. *Essen Bioscience*, 1-5. (2015).
- 10 Langst, G. & Manelyte, L. Chromatin Remodelers: From Function to Dysfunction. *Genes (Basel)* **6**, 299-324, doi:10.3390/genes6020299 (2015).
- 11 O'Connor, L., Gilmour, J. & Bonifer, C. The Role of the Ubiquitously Expressed Transcription Factor Sp1 in Tissue-specific Transcriptional Regulation and in Disease. *Yale J Biol Med* **89**, 513-525 (2016).
- 12 Bhatt, D. & Ghosh, S. Regulation of the NF-κB-Mediated Transcription of Inflammatory Genes. *Frontiers in Immunology* **5**, doi:10.3389/fimmu.2014.00071 (2014).
- 13 Gao, J. *et al.* Endothelial p300 promotes portal hypertension and hepatic fibrosis through CCL2-mediated angiocrine signaling. *Hepatology* **n/a**, doi:<https://doi.org/10.1002/hep.31617> (2020).
- 14 Thakore, P. I. *et al.* Highly specific epigenome editing by CRISPR-Cas9 repressors for silencing of distal regulatory elements. *Nat Methods* **12**, 1143-1149, doi:10.1038/nmeth.3630 (2015).
- 15 Bertola, A., Mathews, S., Ki, S. H., Wang, H. & Gao, B. Mouse model of chronic and binge ethanol feeding (the NIAAA model). *Nature protocols* **8**, 627-637, doi:10.1038/nprot.2013.032 (2013).
- 16 Gao, B., Ahmad, M. F., Nagy, L. E. & Tsukamoto, H. Inflammatory pathways in alcoholic steatohepatitis. *Journal of hepatology* **70**, 249-259, doi:<https://doi.org/10.1016/j.jhep.2018.10.023> (2019).
- 17 Wilkin, R. J. W., Lalor, P. F., Parker, R. & Newsome, P. N. Murine Models of Acute Alcoholic Hepatitis and Their Relevance to Human Disease. *The American Journal of Pathology* **186**, 748-760, doi:<https://doi.org/10.1016/j.ajpath.2015.12.003> (2016).
- 18 Kong, X. *et al.* Activation of autophagy attenuates EtOH-LPS-induced hepatic steatosis and injury through MD2 associated TLR4 signaling. *Scientific Reports* **7**, 9292, doi:10.1038/s41598-017-09045-z (2017).
- 19 Arroyo, V., Moreau, R. & Jalan, R. Acute-on-Chronic Liver Failure. *N Engl J Med* **382**, 2137-2145, doi:10.1056/NEJMra1914900 (2020).
- 20 Hillyer, P., Mordelet, E., Flynn, G. & Male, D. Chemokines, chemokine receptors and adhesion molecules on different human endothelia: discriminating the tissue-specific functions that affect leucocyte migration. *Clin Exp Immunol* **134**, 431-441, doi:10.1111/j.1365-2249.2003.02323.x (2003).
- 21 Poisson, J. *et al.* Liver sinusoidal endothelial cells: Physiology and role in liver diseases. *Journal of hepatology* **66**, 212-227, doi:10.1016/j.jhep.2016.07.009 (2017).
- 22 Lopez, B. G., Tsai, M. S., Baratta, J. L., Longmuir, K. J. & Robertson, R. T. Characterization of Kupffer cells in livers of developing mice. *Comp Hepatol* **10**, 2, doi:10.1186/1476-5926-10-2 (2011).
- 23 Kmiec, Z. Cooperation of liver cells in health and disease. *Adv Anat Embryol Cell Biol* **161**, III-XIII, 1-151, doi:10.1007/978-3-642-56553-3 (2001).

- 24 Brown, Jonathan D. *et al.* NF- κ B Directs Dynamic Super Enhancer Formation in Inflammation and Atherogenesis. *Molecular Cell* **56**, 219-231, doi:<https://doi.org/10.1016/j.molcel.2014.08.024> (2014).
- 25 Dominguez, M. *et al.* Hepatic Expression of CXC Chemokines Predicts Portal Hypertension and Survival in Patients With Alcoholic Hepatitis. *Gastroenterology* **136**, 1639-1650, doi:<https://doi.org/10.1053/j.gastro.2009.01.056> (2009).
- 26 MacParland, S. A. *et al.* Single cell RNA sequencing of human liver reveals distinct intrahepatic macrophage populations. *Nat Commun* **9**, 4383, doi:10.1038/s41467-018-06318-7 (2018).
- 27 Ramachandran, P. *et al.* Resolving the fibrotic niche of human liver cirrhosis at single-cell level. *Nature* **575**, 512-518, doi:10.1038/s41586-019-1631-3 (2019).
- 28 Aizarani, N. *et al.* A human liver cell atlas reveals heterogeneity and epithelial progenitors. *Nature* **572**, 199-204, doi:10.1038/s41586-019-1373-2 (2019).
- 29 Xue, J. *et al.* Transcriptome-based network analysis reveals a spectrum model of human macrophage activation. *Immunity* **40**, 274-288, doi:10.1016/j.immuni.2014.01.006 (2014).
- 30 Cui, H. *et al.* Selective N-Terminal BET Bromodomain Inhibitors by Targeting Non-Conserved Residues and Structured Water Displacement**. *Angewandte Chemie International Edition* **60**, 1220-1226, doi:<https://doi.org/10.1002/anie.202008625> (2021).

Reviewers' Comments:

Reviewer #1:

Remarks to the Author:

The authors have globally addressed my comments.

Regarding the in vivo data, although there are some inconsistencies between the mice models due to differences in severity of the injury, the most reliable one remaining the chronic, alcohol/LPS model, in which steatosis, inflammation and increase in caspase3 is observed. I would therefore move all the data from supplemental figures obtained with this model to the main text

Reviewer #2:

Remarks to the Author:

With the additional experiments presented and discussion of the issues of concern, the authors have adequately addressed my concerns.

Reviewer #3:

Remarks to the Author:

The manuscript by Liu et al has indeed been approved and the authors have addressed most of my comments. However, there are still some statements concerning the source of Cxcl1,6 and 8 expression in human livers that should be modified. Moreover, addition of the new NFkB and H3K27Ac ChIPseq data could be better presented so the reader gets the full overview of the data at the Cxcl locus. See specific comments below.

The authors included scRNAseq data analysis to confirm high expression of Cxcl1,6 and 8 in LSECs. The data from scRNAseq analysis is included in suppl table 1. According to this table LSECs are not the primary source of Cxcl1,6 and 8 expression in the liver and disagrees with the data shown in figure 2a. The scRNAseq data suggests that Cxcl1 and 6 are primarily expressed in cholangiocytes and Cxcl8 is primarily expressed in macrophages. This means that the signal from bulk analysis is not only from LSECs but potentially also cholangiocytes and/or macrophages. Please comment on this and rephrase the statement on line 188-190: "We also analyzed recent single-cell RNA-seq (scRNA-seq) data in human and mouse livers and found LSECs to be among top sources of CXCL chemokine production in multiple studies (Suppl. Table 1)".

In the discussion (line 441) the authors propose that LSECs are the first responders to the inflammatory cascade. However, given the high Cxcl1,6 and 8 expression in cholangiocytes (from the scRNAseq data in suppl table 1) wouldn't it be possible that cholangiocytes are also a part of the first response? This should also be discussed.

Data from a NFkB and H3K27Ac ChIPseq in LSECs is now included in suppl. Figure 11. In line 237-239 the authors now state: "We performed NF-κB (RELA/p65) ChIP-seq in LSECs and observed a similar pattern of enhanced NF-κB binding with TNFα stimulation (Suppl. Figure 11a)". This statement seems inaccurate. Less NF-κB peaks are observed in LSECs compared to HUVEC at regions upstream Cxcl8. Thus, statement should be modified. Also, the authors only show upstream NF-κB binding and H3K27Ac occupancy. To get the full overview they should include the LSEC NF-κB data in figure 10b or at least use the same window size of the genome in figure 11a and figure 10b, so NF-κB binding in LSECs is visualized across the entire Cxcl locus.

Response to Reviewers:

Thank you for your helpful comments. We value your feedback and have attempted to address each question brought to our attention. Please refer to our point-by-point response letter below with references. We hope that our revised contents and responses will satisfy the raised concerns.

Reviewer #1 (Remarks to the Author):

The authors have globally addressed my comments.

Regarding the in vivo data, although there are some inconsistencies between the mice models due to differences in severity of the injury, the most reliable one remaining the chronic, alcohol/LPS model, in which steatosis, inflammation and increase in caspase3 is observed. I would therefore move all the data from supplemental figures obtained with this model to the main text.

We thank the reviewer for his/her suggestion. We have amended main **Figure 6** and **Figure 7** to include **Figure 6d, 6f**, and **Figure 7d** to incorporate results previously presented in Supplementary Materials.

Reviewer #2 (Remarks to the Author):

With the additional experiments presented and discussion of the issues of concern, the authors have adequately addressed my concerns.

We thank the reviewer for the positive comments.

Reviewer #3 (Remarks to the Author):

The manuscript by Liu et al has indeed been approved and the authors have addressed most of my comments. However, there are still some statements concerning the source of Cxcl1,6 and 8 expression in human livers that should be modified. Moreover, addition of the new NFkB and H3K27Ac ChIPseq data could be better presented so the reader gets the full overview of the data at the Cxcl locus. See specific comments below.

We thank the reviewer for these comments. We have amended the manuscript to address these concerns; they are addressed in greater detail in point-by-point responses below.

1/2. The authors included scRNAseq data analysis to confirm high expression of Cxcl1,6 and 8 in LSECs. The data from scRNAseq analysis is included in suppl table 1. According to this table LSECs are not the primary source of Cxcl1,6 and 8 expression in the liver and disagrees with the data shown in figure 2a. The scRNAseq data suggests that Cxcl1 and 6 are primarily expressed in cholangiocytes and Cxcl8 is primarily expressed in macrophages. This means that the signal from bulk analysis is not only from LSECs but potentially also cholangiocytes and/or macrophages. Please comment on this and rephrase the statement on line 188-190: "We also analyzed recent single-cell RNA-seq (scRNA-seq) data in human and mouse livers and found LSECs to be among top sources of CXCL chemokine production in multiple studies (Supp. Table 1)".

In the discussion (line 441) the authors propose that LSECs are the first responders to the inflammatory cascade. However, given the high Cxcl1,6 and 8 expression in cholangiocytes (from the scRNAseq data in suppl table 1) wouldn't it be possible that cholangiocytes are also a part of the first response? This should also be discussed.

We thank the reviewer for these comments. We agree that scRNAseq data highlighted contribution of cholangiocytes and macrophages in addition to LSECs in CXCL production. We acknowledge that multiple cell sources collectively contribute to CXCL upregulation in AH inflammatory response. Ductular reaction cells, which include cholangiocytes and hepatocyte progenitors, have been shown to increase expression of CXCL chemokines in AH¹. We and others have shown that macrophages produce CXCL chemokines after differentiation from monocytes (**Supp. Figure 22**). It is possible that our signal from bulk

RNA-seq and ChIP-seq analysis are not only from LSECs but potentially also cholangiocytes and/or macrophages. We do note the relatively larger liver cell population percentage of LSECs compared to cholangiocytes and/or macrophages, we have rephrased the highlighted statement to be: “We also analyzed recent single-cell RNA-seq (scRNA-seq) data in human and mouse livers. We found cholangiocytes, macrophages and LSECs to be among top sources of CXCL chemokine production in multiple studies (**Supp. Table 1**)^{2,3}” “As LSECs represent a large cell population in liver, we hypothesize that LSECs, as resident liver cells with direct contact with infiltrating immune cells, may be particularly important in the early inflammatory response before the onset of significant immune cell infiltration.” Lines 188-190 and 193-195.

Data from a NF- κ B and H3K27Ac ChIP-seq in LSECs is now included in suppl. Figure 11. In line 237-239 the authors now state: “We performed NF- κ B (RELA/p65) ChIP-seq in LSECs and observed a similar pattern of enhanced NF- κ B binding with TNF α stimulation (Supp. Figure 11a)”. This state seems inaccurate. Less NF- κ B peaks are observed in LSECs compared to HUVEC at regions upstream Cxcl8. Thus, statement should be modified. Also, the authors only show upstream NF- κ B binding and H3K27Ac occupancy. To get the full overview they should include the LSEC NF- κ B data in in figure 10b or at least use the same window size of the genome in figure 11a and figure 10b, so NF- κ B binding in LSECs is visualized across the entire Cxcl locus.

We thank the reviewer for these suggestions. NF- κ B occupancy is enhanced with TNF α stimulation in both LSEC and HUVEC NF- κ B ChIP-seq. LSEC ChIP-seq had lower signal to noise ratio due to lower binding affinity of anti-NF- κ B antibody compared to the previous anti-NF- κ B antibody used in the HUVEC ChIP-seq, which has been discontinued. We have amended **Supp. Figure 11a** to show the full CXCL locus. We have amended the statements indicated above to “We performed NF- κ B (RELA/p65) ChIP-seq in LSECs and also observed enhanced NF- κ B binding with TNF α stimulation (**Supp. Figure 11a**). The discrepancy of the number and sizes of peaks between HUVEC/LSEC ChIP-seq might be due to endothelial cells from different origins or the usage of different anti-NF- κ B antibody in this assay.” Lines 238-242.

References:

- 1 Aguilar-Bravo, B. *et al.* Ductular Reaction Cells Display an Inflammatory Profile and Recruit Neutrophils in Alcoholic Hepatitis. *Hepatology* **69**, 2180-2195, doi:10.1002/hep.30472 (2019).
- 2 Xiong, X. *et al.* Landscape of Intercellular Crosstalk in Healthy and NASH Liver Revealed by Single-Cell Secretome Gene Analysis. *Mol Cell* **75**, 644-660 e645, doi:10.1016/j.molcel.2019.07.028 (2019).
- 3 Aizarani, N. *et al.* A human liver cell atlas reveals heterogeneity and epithelial progenitors. *Nature* **572**, 199-204, doi:10.1038/s41586-019-1373-2 (2019).

Reviewers' Comments:

Reviewer #3:

Remarks to the Author:

The authors have addressed all the comments raised by this reviewer and thus recommends to publish the manuscript.